# A GOOD IMAGE GENERATOR IS WHAT YOU NEED FOR HIGH-RESOLUTION VIDEO SYNTHESIS

**Yu Tian**[1]***,Jian Ren**[2]**, Menglei Chai**[2]**, Kyle Olszewski**[2]**, Xi Peng**[3]**,**
**Dimitris N. Metaxas**[1]**, Sergey Tulyakov**[2]
[1]Rutgers University, [2]Snap Inc., [3]University of Delaware
{yt219, dnm}@cs.rutgers.edu,
{jren, mchai, kolszewski, stulyakov}@snapchat.com

## ABSTRACT

Image and video synthesis are closely related areas aiming at generating content from noise. While rapid progress has been demonstrated in improving image-based models to handle large resolutions, high-quality renderings, and wide variations in image content, achieving comparable video generation results remains problematic. We present a framework that leverages contemporary image generators to render high-resolution videos. We frame the video synthesis problem as discovering a trajectory in the latent space of a *pre-trained* and *fixed* image generator. Not only does such a framework render high-resolution videos, but it also is an order of magnitude more computationally efficient. We introduce a motion generator that discovers the desired trajectory, in which content and motion are disentangled. With such a representation, our framework allows for a broad range of applications, including content and motion manipulation. Furthermore, we introduce a new task, which we call *cross-domain video synthesis*, in which the image and motion generators are trained on disjoint datasets belonging to *different* domains. This allows for generating moving objects for which the desired video data is not available. Extensive experiments on various datasets demonstrate the advantages of our methods over existing video generation techniques. Code will be released at *https://github.com/snap-research/MoCoGAN-HD*.

## 1 INTRODUCTION

Video synthesis seeks to generate a sequence of moving pictures from noise. While its closely related counterpart—image synthesis—has seen substantial advances in recent years, allowing for synthesizing at high resolutions (Karras et al., 2017), rendering images often indistinguishable from real ones (Karras et al., 2019), and supporting multiple classes of image content (Zhang et al., 2019), contemporary improvements in the domain of video synthesis have been comparatively modest. Due to the statistical complexity of videos and larger model sizes, video synthesis produces relatively low-resolution videos, yet requires longer training times. For example, scaling the image generator of Brock et al. (2019) to generate $256 \times 256$ videos requires a substantial computational budget[1]. Can we use a similar method to attain higher resolutions? We believe a different approach is needed.

There are two desired properties for generated videos: (i) high quality for each individual frame, and (ii) the frame sequence should be temporally consistent, *i.e.* depicting the same content with plausible motion. Previous works (Tulyakov et al., 2018; Clark et al., 2019) attempt to achieve both goals with a single framework, making such methods computationally demanding when high resolution is desired. We suggest a different perspective on this problem. We hypothesize that, given an image generator that has learned the distribution of video frames as independent images, a video can be represented as a sequence of latent codes from this generator. The problem of video synthesis can then be framed as discovering a latent trajectory that renders temporally consistent images. Hence, we demonstrate that (i) can be addressed by a *pre-trained* and *fixed* image generator, and (ii) can be achieved using the proposed framework to create appropriate image sequences.

---

*Work done while at Snap Inc.

[1]We estimate that the cost of training a model such as DVD-GAN (Clark et al., 2019) once requires > \$30K.

To discover the appropriate latent trajectory, we introduce a motion generator, implemented via two recurrent neural networks, that operates on the initial content code to obtain the motion representation. We model motion as a residual between continuous latent codes that are passed to the image generator for individual frame generation. Such a residual representation can also facilitate the disentangling of motion and content. The motion generator is trained using the chosen image discriminator with contrastive loss to force the content to be temporally consistent, and a patch-based multi-scale video discriminator for learning motion patterns. Our framework supports contemporary image generators such as StyleGAN2 (Karras et al., 2019) and BigGAN (Brock et al., 2019).

We name our approach as MoCoGAN-HD (Motion and Content decomposed GAN for High-Definition video synthesis) as it features several major advantages over traditional video synthesis pipelines. First, it transcends the limited resolutions of existing techniques, allowing for the generation of high-quality videos at resolutions up to $1024 \times 1024$. Second, as we search for a latent trajectory in an image generator, our method is computationally more efficient, requiring an order of magnitude less training time than previous video-based works (Clark et al., 2019). Third, as the image generator is fixed, it can be trained on a separate high-quality image dataset. Due to the disentangled representation of motion and content, our approach can learn motion from a video dataset and apply it to an image dataset, even in the case of two datasets belonging to *different* domains. It thus unleashes the power of an image generator to synthesize high quality videos when a domain (*e.g.*, dogs) contains many high-quality images but no corresponding high-quality videos (see Fig. 4). In this manner, our method can generate realistic videos of objects it has never seen moving during training (such as generating realistic pet face videos using motions extracted from images of talking people). We refer to this new video generation task as *cross-domain video synthesis*. Finally, we quantitatively and qualitatively evaluate our approach, attaining state-of-the-art performance on each benchmark, and establish a challenging new baseline for video synthesis methods.

## 2 RELATED WORK

**Video Synthesis**. Approaches to image generation and translation using Generative Adversarial Networks (GANs) (Goodfellow et al., 2014) have demonstrated the ability to synthesize high quality images (Radford et al., 2016; Zhang et al., 2019; Brock et al., 2019; Donahue & Simonyan, 2019; Jin et al., 2021). Built upon image translation (Isola et al., 2017; Wang et al., 2018b), works on video-to-video translation (Bansal et al., 2018; Wang et al., 2018a) are capable of converting an input video to a high-resolution output in another domain. However, the task of high-fidelity video generation, in the unconditional setting, is still a difficult and unresolved problem. Without the strong conditional inputs such as segmentation masks (Wang et al., 2019) or human poses (Chan et al., 2019; Ren et al., 2020) that are employed by video-to-video translation works, generating videos following the distribution of training video samples is challenging. Earlier works on GAN-based video modeling, including MDPGAN (Yushchenko et al., 2019), VGAN (Vondrick et al., 2016), TGAN (Saito et al., 2017), MoCoGAN (Tulyakov et al., 2018), ProgressiveVGAN (Acharya et al., 2018), TGANv2 (Saito et al., 2020) show promising results on low-resolution datasets. Recent efforts demonstrate the capacity to generate more realistic videos, but with significantly more computation (Clark et al., 2019; Weissenborn et al., 2020). In this paper, we focus on generating realistic videos using manageable computational resources. LDVDGAN (Kahembwe & Ramamoorthy, 2020) uses low dimensional discriminator to reduce model size and can generate videos with resolution up to $512 \times 512$, while we decrease training cost by utilizing a pre-trained image generator. The high-quality generation is achieved by using pre-trained image generators, while the motion trajectory is modeled within the latent space. Additionally, learning motion in the latent space allows us to easily adapt the video generation model to the task of video prediction (Denton et al., 2017), in which the starting frame is given (Denton & Fergus, 2018; Zhao et al., 2018; Walker et al., 2017; Villegas et al., 2017b;a; Babaeizadeh et al., 2017; Hsieh et al., 2018; Byeon et al., 2018), by inverting the initial frame through the generator (Abdal et al., 2020), instead of training an extra image encoder (Tulyakov et al., 2018; Zhang et al., 2020).

**Interpretable Latent Directions**. The latent space of GANs is known to consist of semantically meaningful vectors for image manipulation. Both supervised methods, either using human annotations or pre-trained image classifiers (Goetschalckx et al., 2019; Shen et al., 2020), and unsupervised methods (Jahanian et al., 2020; Plumerault et al., 2020), are able to find interpretable directions for image editing, such as supervising directions for image rotation or background removal (Voynov &

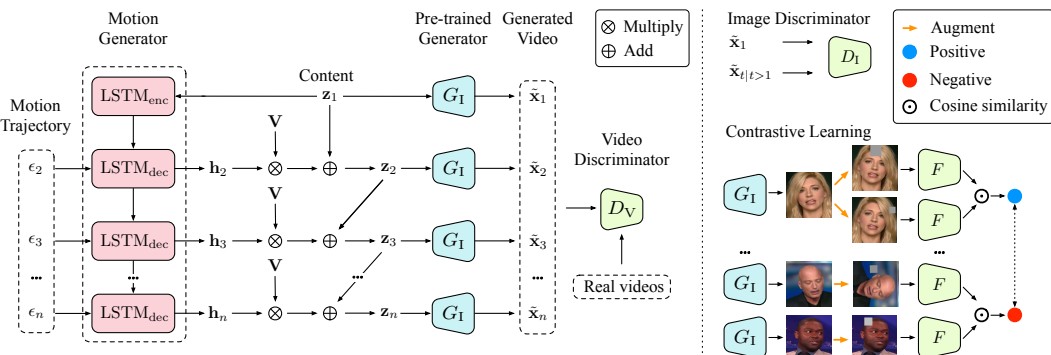

Figure 1: **Left**: Given an initial latent code $\mathbf{z}_1$, a trajectory $\epsilon_t$, and a PCA basis $\mathbf{V}$, the motion generator $G_M$ encodes $\mathbf{z}_1$ using $\mathrm{LSTM}_{\mathrm{enc}}$ to get the initial hidden state and uses $\mathrm{LSTM}_{\mathrm{dec}}$ to estimate hidden states for future frames. The image generator $G_I$ synthesizes images using the predicted latent codes. The discriminator $D_V$ is trained on both real and generated video sequences. **Right**: For each generated video, the first and subsequent frames are sent to an image discriminator $D_I$. An encoder-like network $F$ calculates the features of synthesized images used to compute the contrastive loss $\mathcal{L}_{\mathrm{contr}}$ with positive (same image content, but different augmentation, shown in blue) and negative pairs (different image content and augmentation, shown in red).

Babenko, 2020; Shen & Zhou, 2020). We further consider the motion vectors in the latent space. By disentangling the motion trajectories in an unsupervised fashion, we are able to transfer the motion information from a video dataset to an image dataset in which no temporal information is available.

**Contrastive Representation Learning** is widely studied in unsupervised learning tasks (He et al., 2020; Chen et al., 2020a;b; Hénaff et al., 2020; Löwe et al., 2019; Oord et al., 2018; Misra & Maaten, 2020). Related inputs, such as images (Wu et al., 2018) or latent representations (Hjelm et al., 2019), which can vary while training due to data augmentation, are forced to be close by minimizing differences in their representation during training. Recent work (Park et al., 2020) applies noise-contrastive estimation (Gutmann & Hyvärinen, 2010) to image generation tasks by learning the correspondence between image patches, achieving performance superior to that attained when using cycle-consistency constraints (Zhu et al., 2017; Yi et al., 2017). On the other hand, we learn an image discriminator to create videos with coherent content by leveraging contrastive loss (Hadsell et al., 2006) along with an adversarial loss (Goodfellow et al., 2014).

## 3 METHOD

In this section, we introduce our method for high-resolution video generation. Our framework is built on top of a *pre-trained* image generator (Karras et al., 2020a;b; Zhao et al., 2020a;b), which helps to generate high-quality image frames and boosts the training efficiency with manageable computational resources. In addition, with the image generator fixed during training, we can disentangle video motion from image content, and enable video synthesis even when the image content and the video motion come from different domains.

More specifically, our inference framework includes a motion generator $G_M$ and an image generator $G_I$. $G_M$ is implemented with two LSTM networks (Hochreiter & Schmidhuber, 1997) and predicts the latent motion *trajectory* $\mathbf{Z} = \{\mathbf{z}_1, \mathbf{z}_2, \cdots, \mathbf{z}_n\}$, where $n$ is the number of frames in the synthesized video. The image generator $G_I$ can thus synthesize each individual frame from the motion trajectory. The generated video sequence $\tilde{\mathbf{v}}$ is given by $\tilde{\mathbf{v}} = \{\tilde{\mathbf{x}}_1, \tilde{\mathbf{x}}_2, \cdots, \tilde{\mathbf{x}}_n\}$. For each synthesized frame $\tilde{\mathbf{x}}_t$, we have $\tilde{\mathbf{x}}_t = G_I(\mathbf{z}_t)$ for $t = 1, 2, \cdots, n$. We also define the real video clip as $\mathbf{v} = \{\mathbf{x}_1, \mathbf{x}_2, \cdots, \mathbf{x}_n\}$ and the training video distribution as $p_v$.

To train the motion generator $G_M$ to discover the desired motion trajectory, we apply a video discriminator to constrain the generated motion patterns to be similar to those of the training videos, and an image discriminator to force the frame content to be temporally consistent. Our framework is illustrated in Fig. 1. We describe each component in more detail in the following sections.

### 3.1 Motion Generator

The motion generator $G_{\mathrm{M}}$ predicts consecutive latent codes using an input code $\mathbf{z}_1 \in \mathcal{Z}$, where the latent space $\mathcal{Z}$ is also shared by the image generator. For BigGAN (Brock et al., 2019), we sample $\mathbf{z}_1$ from the normal distribution $p_z$. For StyleGAN2 (Karras et al., 2020b), $p_z$ is the distribution after the multi-layer perceptron (MLP), as the latent codes within this distribution can be semantically disentangled better than when using the normal distribution (Shen et al., 2020; Zhu et al., 2020).

Formally, $G_{\mathrm{M}}$ includes an LSTM encoder $\mathrm{LSTM}_{\mathrm{enc}}$, which encodes $\mathbf{z}_1$ to get the initial hidden state, and a LSTM decoder $\mathrm{LSTM}_{\mathrm{dec}}$, which estimates $n-1$ continuous hidden states recursively:

$$
\begin{aligned}
\mathbf{h}_1, \mathbf{c}_1 &= \mathrm{LSTM}_{\mathrm{enc}}(\mathbf{z}_1), \\
\mathbf{h}_t, \mathbf{c}_t &= \mathrm{LSTM}_{\mathrm{dec}}(\epsilon_t, (\mathbf{h}_{t-1}, \mathbf{c}_{t-1})), \quad t = 2, 3, \cdots, n,
\end{aligned}
\tag{1}
$$

where $\mathbf{h}$ and $\mathbf{c}$ denote the hidden state and cell state respectively, and $\epsilon_t$ is a noise vector sampled from the normal distribution to model the motion diversity at timestamp $t$.

**Motion Disentanglement.** Prior work (Tulyakov et al., 2018) applies $\mathbf{h}_t$ as the motion code for the frame to be generated, while the content code is fixed for all frames. However, such a design requires a recurrent network to estimate the motion while preserving consistent content from the latent vector, which is difficult to learn in practice. Instead, we propose to use a sequence of motion residuals for estimating the motion trajectory. Specifically, we model the motion residual as the linear combination of a set of interpretable directions in the latent space (Shen & Zhou, 2020; Härkönen et al., 2020). We first conduct principal component analysis (PCA) on $m$ randomly sampled latent vectors from $\mathcal{Z}$ to get the basis $\mathbf{V}$. Then, we estimate the motion direction from the previous frame $\mathbf{z}_{t-1}$ to the current frame $\mathbf{z}_t$ by using $\mathbf{h}_t$ and $\mathbf{V}$ as follows:

$$
\mathbf{z}_t = \mathbf{z}_{t-1} + \lambda \cdot \mathbf{h}_t \cdot \mathbf{V}, \quad t = 2, 3, \cdots, n,
\tag{2}
$$

where the hidden state $\mathbf{h}_t \in [-1, 1]$, and $\lambda$ controls the step given by the residual. With Eqn. 1 and Eqn. 2, we have $G_{\mathrm{M}}(\mathbf{z}_1) = \{\mathbf{z}_1, \mathbf{z}_2, \cdots, \mathbf{z}_n\}$, and the generated video $\tilde{\mathbf{v}}$ is given as $\tilde{\mathbf{v}} = G_{\mathrm{I}}(G_{\mathrm{M}}(\mathbf{z}_1))$.

**Motion Diversity.** In Eqn. 1, we introduce a noise vector $\epsilon_t$ to control the diversity of motion. However, we observe that the LSTM decoder tends to neglect the $\epsilon_t$, resulting in *motion mode collapse*, meaning that $G_{\mathrm{M}}$ cannot capture the diverse motion patterns from training videos and generate distinct videos from one initial latent code with similar motion patterns for different sequences of noise vectors. To alleviate this issue, we introduce a mutual information loss $\mathcal{L}_{\mathrm{m}}$ to maximize the *mutual information* between the hidden vector $\mathbf{h}_t$ and the noise vector $\epsilon_t$. With $\mathrm{sim}(\mathbf{u}, \mathbf{v}) = \mathbf{u}^T \mathbf{v} / \|\mathbf{u}\| \|\mathbf{v}\|$ denoting the cosine similarity between vectors $\mathbf{u}$ and $\mathbf{v}$, we define $\mathcal{L}_{\mathrm{m}}$ as follows:

$$
\mathcal{L}_{\mathrm{m}} = \frac{1}{n-1} \sum_{t=2}^{n} \mathrm{sim}(H(\mathbf{h}_t), \epsilon_t),
\tag{3}
$$

where $H$ is a 2-layer MLP that serves as a mapping function.

**Learning.** To learn the appropriate parameters for the motion generator $G_{\mathrm{M}}$, we apply a multi-scale video discriminator $D_{\mathrm{V}}$ to tell whether a video sequence is real or synthesized. The discriminator is based on the architecture of PatchGAN (Isola et al., 2017). However, we use 3D convolutional layers in $D_{\mathrm{V}}$, as they can model temporal dynamics better than 2D convolutional layers. We divide input video sequence into small 3D patches, and classify each patch as real or fake. The local responses for the input sequence are averaged to produce the final output. Additionally, each frame in the input video sequence is conditioned on the first frame, as it falls into the distribution of the pre-trained image generator, for more stable training. We thus optimize the following adversarial loss to learn $G_{\mathrm{M}}$ and $D_{\mathrm{V}}$:

$$
\mathcal{L}_{D_{\mathrm{V}}} = \mathbb{E}_{\mathbf{v} \sim p_v} \left[ \log D_{\mathrm{V}}(\mathbf{v}) \right] + \mathbb{E}_{\mathbf{z}_1 \sim p_z} \left[ \log(1 - D_{\mathrm{V}}(G_{\mathrm{I}}(G_{\mathrm{M}}(\mathbf{z}_1)))) \right].
\tag{4}
$$

### 3.2 Contrastive Image Discriminator

As our image generator is pre-trained, we may use an image generator that is trained on a given domain, *e.g.* images of animal faces (Choi et al., 2020), and learn the motion generator parameters using videos from a different domain, such as videos of human facial expressions (Nagrani et al.,

2017). With Eqn. 4 alone, however, we lack the ability to explicitly constrain the generated images $\tilde{\mathbf{x}}_{t|t>1}$ to possess similar quality and content as the first image $\tilde{\mathbf{x}}_1$, which is sampled from the image space of the image generator and thus has high fidelity. Hence, we introduce a contrastive image discriminator $D_{\mathrm{I}}$, which is illustrated in Fig. 1, to match both image quality and content between $\tilde{\mathbf{x}}_1$ and $\tilde{\mathbf{x}}_{t|t>1}$.

**Quality Matching.** To increase the perceptual quality, we train $D_{\mathrm{I}}$ and $G_{\mathrm{M}}$ adversarially by forwarding $\tilde{\mathbf{x}}_t$ into the discriminator $D_{\mathrm{I}}$ and using $\tilde{\mathbf{x}}_1$ as real sample and $\tilde{\mathbf{x}}_{t|t>1}$ as the fake sample.

$$\mathcal{L}_{D_{\mathrm{I}}} = \mathbb{E}_{\mathbf{z}_1 \sim p_z} \left[ \log D_{\mathrm{I}}(G_{\mathrm{I}}(\mathbf{z}_1)) \right] + \mathbb{E}_{\mathbf{z}_1 \sim p_z, \mathbf{z}_t \sim G_{\mathrm{M}}(\mathbf{z}_1)|t>1} \left[ \log(1 - D_{\mathrm{I}}(G_{\mathrm{I}}(\mathbf{z}_t))) \right]. \tag{5}$$

**Content Matching.** To learn content similarity between frames within a video, we use the image discriminator as a feature extractor and train it with a form of contrastive loss known as InfoNCE (Oord et al., 2018). The goal is that pairs of images with the same content should be close together in embedding space, while images containing different content should be far apart.

Given a minibatch of $N$ generated videos $\{\tilde{\mathbf{v}}^{(1)}, \tilde{\mathbf{v}}^{(2)}, \cdots, \tilde{\mathbf{v}}^{(N)}\}$, we randomly sample one frame $t$ from each video: $\{\tilde{\mathbf{x}}_t^{(1)}, \tilde{\mathbf{x}}_t^{(2)}, \cdots, \tilde{\mathbf{x}}_t^{(N)}\}$, and make two randomly augmented versions $(\tilde{\mathbf{x}}_t^{(ia)}, \tilde{\mathbf{x}}_t^{(ib)})$ for each frame $\tilde{\mathbf{x}}_t^{(i)}$, resulting in $2N$ samples. $(\tilde{\mathbf{x}}_t^{(ia)}, \tilde{\mathbf{x}}_t^{(ib)})$ are positive pairs, as they share the same content. $(\tilde{\mathbf{x}}_t^{(i\cdot)}, \tilde{\mathbf{x}}_t^{(j\cdot)})$ are all negative pairs for $i \neq j$.

Let $F$ be an encoder network, which shares the same weights and architecture of $D_{\mathrm{I}}$, but excluding the last layer of $D_{\mathrm{I}}$ and including a 2-layer MLP as a projection head that produces the representation of the input images. We have a contrastive loss function $\mathcal{L}_{\mathrm{contr}}$, which is the cross-entropy computed across $2N$ augmentations as follows:

$$\mathcal{L}_{\mathrm{contr}} = -\sum_{i=1}^{N} \sum_{\alpha=a}^{b} \log \frac{\exp(\mathrm{sim}(F(\tilde{\mathbf{x}}_t^{(ia)}), F(\tilde{\mathbf{x}}_t^{(ib)}))/\tau)}{\sum_{j=1}^{N} \sum_{\beta=a}^{b} \mathbb{1}_{[j \neq i]}(\exp(\mathrm{sim}(F(\tilde{\mathbf{x}}_t^{(i\alpha)}), F(\tilde{\mathbf{x}}_t^{(j\beta)}))/\tau)}, \tag{6}$$

where $\mathrm{sim}(\cdot, \cdot)$ is the cosine similarity function defined in Eqn. 3, $\mathbb{1}_{[j \neq i]} \in \{0, 1\}$ is equal to 1 iff $j \neq i$, and $\tau$ is a temperature parameter empirically set to $0.07$. We use a momentum decoder mechanism similar to that of MoCo (He et al., 2020) by maintaining a memory bank to delete the oldest negative pairs and update the new negative pairs. We apply augmentation methods including translation, color jittering, and cutout (DeVries & Taylor, 2017) on synthesized images. With the positive and negative pairs generated on-the-fly during training, the discriminator can effectively focus on the content of the input samples.

The choice of positive pairs in Eqn. 6 is specifically designed for cross-domain video synthesis, as videos of arbitrary content from the image domain is not available. In the case that images and videos are from the same domain, the positive and negative pairs are easier to obtain. We randomly select and augment two frames from a real video to create positive pairs sharing the same content, while the negative pairs contain augmented images from different real videos.

Aside from $\mathcal{L}_{\mathrm{contr}}$, we also adopt the feature matching loss (Wang et al., 2018b) $\mathcal{L}_{\mathrm{f}}$ between the generated first frame and other frames by changing the $L_1$ regularization to cosine similarity.

**Full Objective.** The overall loss function for training motion generator, video discriminator, and image discriminator is thus defined as:

$$\min_{G_{\mathrm{M}}}(\max_{D_{\mathrm{V}}} \mathcal{L}_{D_{\mathrm{V}}} + \max_{D_{\mathrm{I}}} \mathcal{L}_{D_{\mathrm{I}}}) + \max_{G_{\mathrm{M}}}(\lambda_{\mathrm{m}} \mathcal{L}_{\mathrm{m}} + \lambda_{\mathrm{f}} \mathcal{L}_{\mathrm{f}}) + \min_{D_{\mathrm{I}}}(\lambda_{\mathrm{contr}} \mathcal{L}_{\mathrm{contr}}) \tag{7}$$

where $\lambda_{\mathrm{m}}$, $\lambda_{\mathrm{contr}}$, and $\lambda_{\mathrm{f}}$ are hyperparameters to balance losses.

## 4 EXPERIMENTS

In this section, we evaluate the proposed approach on several benchmark datasets for video generation. We also demonstrate cross-domain video synthesis for various image and video datasets.

### 4.1 VIDEO GENERATION

We conduct experiments on three datasets including UCF-101 (Soomro et al., 2012), FaceForensics (Rössler et al., 2018), and Sky Time-lapse (Xiong et al., 2018) for unconditional video synthesis. We use StyleGAN2 as the image generator. Training details can be found in Appx. B.

Table 1: IS and FVD on UCF-101.

| Method | IS (↑) | FVD (↓) |
|---|---|---|
| VGAN | 8.31 ± .09 | - |
| TGAN | 11.85 ± .07 | - |
| MoCoGAN | 12.42 ± .07 | - |
| ProgressiveVGAN | 14.56 ± .05 | - |
| LDVD-GAN | 22.91 ± .19 | - |
| TGANv2 | 26.60 ± .47 | 1209 ± 28 |
| DVD-GAN | 27.38 ± .53 | - |
| Ours | **33.95 ± .25** | **700 ± 24** |

Table 2: FVD, ACD, and Human Preference on FaceForensics.

| Method | FVD (↓) | ACD (↓) |
|---|---|---|
| GT | 9.02 | 0.2935 |
| TGANv2 | 58.03 | 0.4914 |
| Ours | **53.26** | **0.3300** |

| Method | Human Preference (%) |
|---|---|
| Ours / TGANv2 | **73.6** / 26.4 |

**UCF-101** is widely used in video generation. The dataset includes $13,320$ videos of $101$ sport categories. The resolution of each video is $320 \times 240$. To process the data, we crop a rectangle with size of $240 \times 240$ from each frame in a video and resize it to $256 \times 256$. We train the motion generator to predict 16 frames. For evaluation, we report Inception Score (IS) (Saito et al., 2020) on $10,000$ generated videos and Fréchet Video Distance (FVD) (Unterthiner et al., 2018) on $2,048$ videos. The classifier used to calculate IS is a C3D network (Tran et al., 2015) that is trained on the Sports-1M dataset (Karpathy et al., 2014) and fine-tuned on UCF-101, which is the same model used in previous works (Saito et al., 2020; Clark et al., 2019).

The quantitative results are shown in Tab. 1. Our method achieves state-of-the-art results for both IS and FVD, and outperforms existing works by a large margin. Interestingly, this result indicates that a well-trained image generator has learned to represent rich motion patterns, and therefore can be used to synthesize high-quality videos when used with a well-trained motion generator.

**FaceForensics** is a dataset containing news videos featuring various reporters. We use all the images from $704$ training videos, with a resolution of $256 \times 256$, to learn an image generator, and sequences of 16 consecutive frames to train motion generator. Note that our network can generate even longer continuous sequences, *e.g.* 64 frames (Fig. 12 in Appx.), though only 16 frames are used for training.

We show the FVD between generated and real video clips (16 frames in length) for different methods in Tab. 2. Additionally, we use the Average Content Distance (ACD) from MoCoGAN (Tulyakov et al., 2018) to evaluate the identity consistency for these human face videos. We calculate ACD values over 256 videos. We also report the two metrics for ground truth (GT) videos. To get FVD of GT videos, we randomly sample two groups of real videos and compute the score. Our method achieves better results than TGANv2 (Saito et al., 2020). Both methods have low FVD values, and can generate complex motion patterns close to the real data. However, the much lower

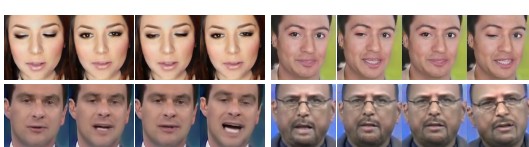

Figure 2: Example generated videos from a model trained on FaceForensics. We can generate natural and photo-realistic videos with various motion patterns, such as eye blink and talking. Four examples show frames 2, 7, 11, and 16.

ACD value of our approach, which is close to GT, demonstrates that the videos it synthesizes have much better identity consistency than the videos from TGANv2. Qualitative examples in Fig. 2 illustrate different motions patterns learned from the dataset. Furthermore, we perform perceptual experiments using Amazon Mechanical Turk (AMT) by presenting a pair of videos from the two methods to users and asking them to select a more realistic one. Results in Tab. 2 indicate our method outperforms TGANv2 in 73.6% of the comparisons.

**Sky Time-Lapse** is a video dataset consisting of dynamic sky scenes, such as moving clouds. The number of video clips for training and testing is $35,392$ and $2,815$, respectively. We resize images to $128 \times 128$ and train the model to generate 16 frames. We compare our methods with the two recent approaches of MDGAN (Xiong et al., 2018) and DTVNet (Zhang et al., 2020), which are specifically designed for this dataset. In Tab. 3, we report the FVD for all three methods. It is clear that our approach significantly outperforms the others. Example sequences are shown in Fig. 3.

Following DTVNet (Zhang et al., 2020), we evaluate the proposed model for the task of *video prediction*. We use the Peak Signal-to-Noise Ratio (PSNR) and Structural Similarity (SSIM) (Wang et al., 2004) as evaluation metrics to measure the frame quality at the pixel level and the structural similarity between synthesized and real video frames. Evaluation is performed on the testing set. We select the first frame $\mathbf{x}_1$ from each video clip and project it to the latent space of the image generator (Abdal et al., 2020) to get $\hat{\mathbf{z}}_1$. We use $\hat{\mathbf{z}}_1$ as the starting latent code for motion generator to get 16 latent codes, and interpolate them to get 32 latent codes to synthesize a video sequence, where the first frame is given by $G_\text{I}(\hat{\mathbf{z}}_1)$. For a fair comparison, we also use $G_\text{I}(\hat{\mathbf{z}}_1)$ as the starting frame for MDGAN and DTVNet to calculate the metrics with ground truth videos. In addition, we calculate the PSNR and SSIM between $\mathbf{x}_1$ and $G_\text{I}(\hat{\mathbf{z}}_1)$ as the upper bound for all methods, which we denote as *Up-B*. Tab. 3 shows the video prediction results, which demonstrate that our method's performance is superior to those of MDGAN and DTVNet. Interestingly, by simply interpolating the motion trajectory, we can easily generate longer video sequence, *e.g.* from 16 to 32 frames, while retaining high quality.

Table 3: Evaluation on Sky Time-lapse for video synthesis and prediction.

| Method | FVD ($\downarrow$) | PSNR ($\uparrow$) | SSIM ($\uparrow$) |
|---|---|---|---|
| Up-B | - | 25.367 | 0.781 |
| MDGAN | 840.95 | 13.840 | 0.581 |
| DTVNet | 451.14 | 21.953 | 0.531 |
| Ours | **77.77** | **22.286** | **0.688** |

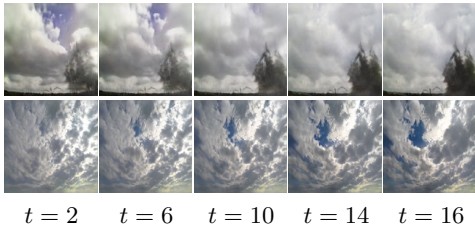

$t = 2 \quad t = 6 \quad t = 10 \quad t = 14 \quad t = 16$

Figure 3: Sample generated frames at several time steps ($t$) for the Sky Time-lapse dataset.

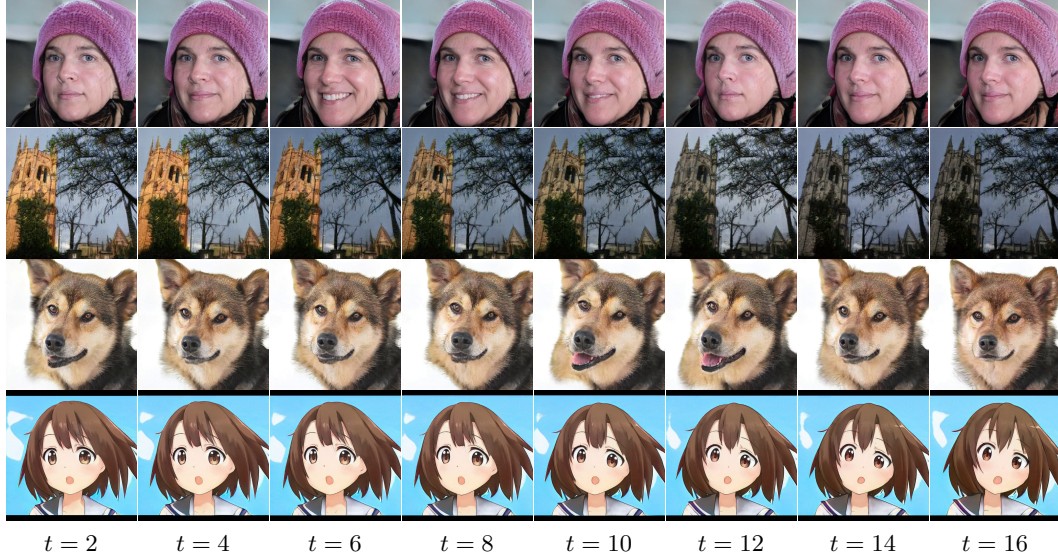

$t = 2 \qquad t = 4 \qquad t = 6 \qquad t = 8 \qquad t = 10 \qquad t = 12 \qquad t = 14 \qquad t = 16$

Figure 4: Example sequences for cross-domain video generation. First Row: (FFHQ, VoxCeleb). Second Row: (LSUN-Church, TLVDB). Third Row: (AFHQ-Dog, VoxCeleb). Fourth Row: (AnimeFaces, VoxCeleb). Images in the first and second rows have a resolution of $256 \times 256$, while the third and fourth rows have a resolution of $512 \times 512$.

## 4.2 CROSS-DOMAIN VIDEO GENERATION

To demonstrate how our approach can disentangle motion from image content and transfer motion patterns from one domain to another, we perform several experiments on various datasets. More specifically, we use the StyleGAN2 model, pre-trained on the FFHQ (Karras et al., 2019), AFHQ-Dog (Choi et al., 2020), AnimeFaces (Branwen, 2019), and LSUN-Church (Yu et al., 2015) datasets, as the image generators. We learn human facial motion from VoxCeleb (Nagrani et al., 2020) and

time-lapse transitions in outdoor scenes from TLVDB (Shih et al., 2013). In these experiments, a pair such as (FFHQ, VoxCeleb) indicates that we synthesize videos with image content from FFHQ and motion patterns from VoxCeleb. We generate videos with a resolution of $256 \times 256$ and $1024 \times 1024$ for FFHQ, $512 \times 512$ for AFHQ-Dog and AnimeFaces, and $256 \times 256$ for LSUN-Church. Qualitative examples for (FFHQ, VoxCeleb), (LSUN-Church, TLVDB), (AFHQ-Dog, VoxCeleb), and (Anime-Faces, VoxCeleb) are shown in Fig. 4, depicting high-quality and temporally consistent videos (more videos, including results with BigGAN as the image generator, are shown in the Appendix).

We also demonstrate how the motion and content are disentangled in Fig. 5 and Fig. 6, which portray generated videos with the same identity but performing diverse motion patterns, and the same motion applied to different identities, respectively. We show results from (AFHQ-Dog, VoxCeleb) (first two rows) and (AnimeFaces, VoxCeleb) (last two rows) in these two figures.

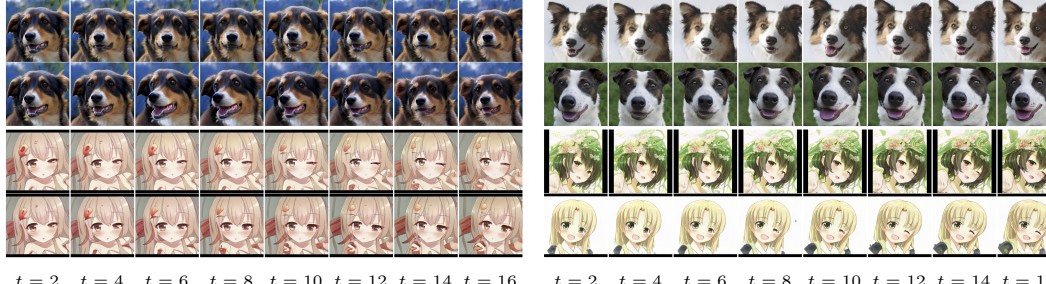

$t=2 \quad t=4 \quad t=6 \quad t=8 \quad t=10 \quad t=12 \quad t=14 \quad t=16$     $t=2 \quad t=4 \quad t=6 \quad t=8 \quad t=10 \quad t=12 \quad t=14 \quad t=16$

Figure 5: The first and second row (also the third and fourth row) share the same initial content code but with different motion codes.

Figure 6: The first and second row (also the third and fourth row) share the same motion code but with different content codes.

### 4.3 ABLATION ANALYSIS

We first report IS and FVD in Tab. 4 for UCF-101 using the following methods: *w/o Eqn. 2* uses $\mathbf{z}_t = \mathbf{h}_t$ instead of estimating the residual as in Eqn. 2; *w/o $D_{\mathrm{I}}$* omits the contrastive image discriminator $D_{\mathrm{I}}$ and uses the video discriminator $D_{\mathrm{V}}$ only for learning the motion generator; *w/o $D_{\mathrm{V}}$* omits $D_{\mathrm{V}}$ during training; and *Full-128* and *Full-256* indicate that we generate videos using our full method with resolutions of $128 \times 128$ and $256 \times 256$, respectively. We resize frames for all methods to $128 \times 128$ when calculating IS and FVD. The full method outperforms all others, proving the importance of each module for learning temporally consistent and high-quality videos.

We perform further analysis of our cross-domain video generation on (FFHQ, VoxCeleb). We compare our full method (*Full*) with two variants. *w/o $\mathcal{L}_{\mathrm{contr}}$* denotes that we omit the contrastive loss (Eqn. 6) from $D_{\mathrm{I}}$, and *w/o $\mathcal{L}_{\mathrm{m}}$* indicates that we omit the mutual information loss (Eqn. 3) for the motion generator. The results in Tab. 5 demonstrate that $\mathcal{L}_{\mathrm{contr}}$ is beneficial for learning videos with coherent content, as employing $\mathcal{L}_{\mathrm{contr}}$ results in lower ACD values and higher human preferences. $\mathcal{L}_{\mathrm{m}}$ also contributes to generating higher quality videos by mitigating motion synchronization. To validate the motion diversity, we show pairs of 9 randomly generated videos from the two methods to users and ask them to choose which one has superior motion diversity, including rotations and facial expressions. User preference suggests that using $\mathcal{L}_{\mathrm{m}}$ increases motion diversity.

Table 4: Ablation study on UCF-101.

| Method | IS ($\uparrow$) | FVD ($\downarrow$) |
|---|---|---|
| w/o Eqn. 2 | 28.20 | 790.87 |
| w/o $D_{\mathrm{I}}$ | 33.22 | 796.67 |
| w/o $D_{\mathrm{V}}$ | 33.84 | 867.43 |
| Full-128 | 32.36 | 838.09 |
| Full-256 | **33.95** | **700.00** |

Table 5: Ablation study on (FFHQ, VoxCeleb).

| Method | w/o $\mathcal{L}_{\mathrm{contr}}$ | w/o $\mathcal{L}_{\mathrm{m}}$ | Full |
|---|---|---|---|
| ACD ($\downarrow$) | 0.5328 | 0.5158 | **0.4353** |

| Method | Human Preference (%) |
|---|---|
| Full *vs* w/o $\mathcal{L}_{\mathrm{contr}}$ | **68.3** / 31.7 |
| Full *vs* w/o $\mathcal{L}_{\mathrm{m}}$ | **64.4** / 35.6 |

### 4.4 LONG SEQUENCE GENERATION

Due to the limitation of computational resources, we train MoCoGAN-HD to synthesize 16 consecutive frames. However, we can generate longer video sequences during inference by applying the following two ways.

**Motion Generator Unrolling.** For motion generator, we can run the LSTM decoder for more steps to synthesize long video sequences. In Fig. 7, we show a synthesized video example of 64 frames using the model trained on the FaceForensics dataset. Our method is capable to synthesize videos with more frames than the number of frames used for training.

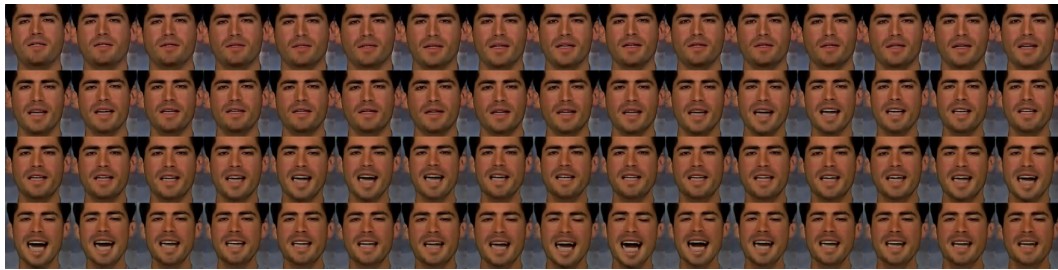

Figure 7: The generation of a 64-frame video using a model trained with 16-frame on FaceForensics.

**Motion Interpolation.** We can do interpolation on the motion trajectory directly to synthesize long videos. Fig. 8 shows an interpolation example of 32-frame on (AFHQ-Dog, VoxCeleb) dataset.

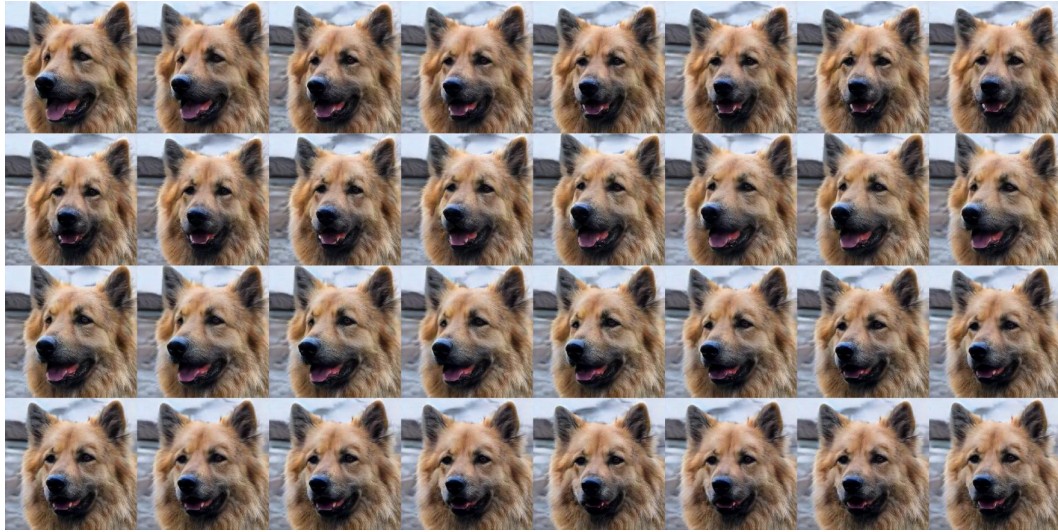

Figure 8: The generation of a 32-frame video on (AFHQ-Dog, VoxCeleb) by doing the interpolation on motion trajectory.

## 5 CONCLUSION

In this work, we present a novel approach to video synthesis. Building on contemporary advances in image synthesis, we show that a good image generator and our framework are essential ingredients to boost video synthesis fidelity and resolution. The key is to find a meaningful trajectory in the image generator's latent space. This is achieved using the proposed motion generator, which produces a sequence of motion residuals, with the contrastive image discriminator and video discriminator. This disentangled representation further extends applications of video synthesis to content and motion manipulation and cross-domain video synthesis. The framework achieves superior results on a variety of benchmarks and reaches resolutions unattainable by prior state-of-the-art techniques.

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

# A    ADDITIONAL DETAILS FOR THE FRAMEWORK

## A.1    ADDITIONAL DETAILS FOR THE MOTION GENERATOR

To use StyleGAN2 (Karras et al., 2020b) as the image generator, we randomly sample $1,000,000$ latent codes from the input space $\mathcal{Z}$ and send them to the 8-layer MLPs to get the latent codes in the space of $\mathcal{W}$. Each latent code is a $512$-dimension vector. We perform PCA on these $1,000,000$ latent codes and select the top $384$ principal components to form the matrix $\mathbf{V} \in \mathbb{R}^{384 \times 512}$, which is used to model the motion residuals in Eqn. 2. The LSTM encoder and the LSTM decoder in the motion generator both have an input size of $512$ and a hidden size of $384$. The noise vector $\epsilon_t$ in Eqn. 1 is also a $512$-dimension vector, and the network $H$ in Eqn. 3 is a 2-layer MLPs with $512$ hidden units in each of the two fully-connected layers.

For BigGAN (Brock et al., 2019), we sample the latent code directly from the space of $\mathcal{Z}$.

## A.2    ADDITIONAL DETAILS FOR THE DISCRIMINATORS

### A.2.1    VIDEO DISCRIMINATOR

The input images for the video discriminator $D_V$ are processed at two scales. We downsample the output images from the image generator to the resolution of $128 \times 128$ and $64 \times 64$. For in-domain video synthesis, the input sequences for $D_V$ have the shape of $6 \times (n-1) \times 128 \times 128$ and $6 \times (n-1) \times 64 \times 64$, where $n$ is the sequence length used for training. For each of the $(n-1)$ subsequent frames, we concatenate the RGB channels of both the first frame and that subsequent frame, resulting in a 6-channel input. For cross-domain video synthesis, the input sequences for $D_V$ have the shape of $3 \times n \times 128 \times 128$ and $3 \times n \times 64 \times 64$, as the concatenation of the first frame will make the discriminator aware the domain gaps. Details for $D_V$ are shown in Tab. 6.

Table 6: The network architecture for video discriminator.

| Operation | Kernel | Strides | # Channels | Norm Type | Nonlinearity |
|-----------|--------|---------|------------|-----------|--------------|
| Conv3d | 4×4 | 2 | 64 | - | Leaky ReLU (0.2) |
| Conv3d | 4×4 | 2 | 128 | InstanceNorm3d | Leaky ReLU (0.2) |
| Conv3d | 4×4 | 2 | 256 | InstanceNorm3d | Leaky ReLU (0.2) |
| Conv3d | 4×4 | 1 | 512 | InstanceNorm3d | Leaky ReLU (0.2) |
| Conv3d | 4×4 | 1 | 1 | - | - |

### A.2.2    IMAGE DISCRIMINATOR

The image discriminator $D_I$ has an architecture based on that of the BigGAN discriminator, except that we remove the self-attention layer. The feature extractor $F$ used for contrastive learning has the same architecture as $D_I$, except that it does not include the last layer of $D_I$ but has two additional fully connected (FC) layers as the projection head. The number of hidden units for these two FC layers are both $256$.

Here we describe in more detail the image augmentation and memory bank techniques used for conducting contrastive learning.

**Image Augmentation.** We perform data augmentation on images to create positive and negative pairs. We normalize the images to $[-1, 1]$ and apply the following augmentation techniques.

- **Affine.** We augment each image with an affine transformation defined with three random parameters: rotation $\alpha_r \in \mathcal{U}(-180, 180)$, translation $\alpha_t \in \mathcal{U}(-0.1, 0.1)$, and scale $\alpha_s \in \mathcal{U}(0.95, 1.05)$.

- **Brightness.** We add a random value $\alpha_b \sim \mathcal{U}(-0.5, 0.5)$ to all channels of each image.

- **Color.** We add a random value $\alpha_c \sim \mathcal{U}(-0.5, 0.5)$ to one randomly-selected channel of each image.

- **Cutout** (DeVries & Taylor, 2017). We mask out pixels in a random subregion of each image to 0. Each subregion starts at a random point and with size $(\alpha_m H, \alpha_m W)$, where $\alpha_m \sim \mathcal{U}(0, 0.25)$ and $(H, W)$ is the image resolution.
- **Flipping**. We horizontally flip the image with the probability of 0.5.

**Memory Bank.** It has been shown that contrastive learning benefits from large batch-sizes and negative pairs (Chen et al., 2020b). To increase the number of negative pairs, we incorporate the memory mechanism from MoCo (He et al., 2020), which designates a memory bank to store negative examples. More specifically, we keep an exponential moving average of the image discriminator, and its output of *fake* video frames are buffered as negative examples. We use a memory bank with a dictionary size of $4,096$.

## B   MORE DETAILS FOR EXPERIMENTS

**Image Generators.** We train the unconditional StyleGAN2 models from scratch on the UCF-101, FaceForensics, Sky Time-lapse, and AFHQ-Dog datasets. We train the image generators with the official Tensorflow code[2] and select the checkpoints that obtain the best Fréchet inception distance (FID) (Heusel et al., 2017) score to be used as the image generators. The FID score of each image generator is shown in Table 7. For FFHQ, AnimeFaces, and LSUN-Church, we simply use the released pre-trained models.

We also train an unconditional BigGAN model on the FFHQ dataset using the public PyTorch code[3]. We train a model with resolution $128 \times 128$ and select the last checkpoint as the image generator.

Table 7: FID of our trained StyleGAN2 models on different datasets.

|     | UCF-101 | FaceForensics | Sky Time-lapse | AFHQ-Dog |
| --- | --- | --- | --- | --- |
| FID | 45.63 | 10.99 | 10.80 | 7.85 |

**Training Time.** We train each image generator for UCF-101, FaceForensics, Sky Time-lapse, and AFHQ-Dog in less than 2 days using 8 Tesla V100 GPUs. For FFHQ, AnimeFaces, and LSUN-Church, we use the released models with no training cost. The training time for video generators ranges from $1.5 \sim 3$ days depending on the datasets (Due to the memory issue, the training for generating videos with resolution of $1,024 \times 1,024$ was done on 8 Quadro RTX 8000, with 5 days). The total training time for all the datasets is $1.5 \sim 5$ days and the estimated cost for training on Google Cloud is \$0.7K~\$2.3K.

**Implementation Details.** We implement our experiments with PyTorch 1.3.1 and also tested them with PyTorch 1.6. We use the Adam optimizer (Kingma & Ba, 2014) with a learning rate of $0.0001$ for $G_M$, $D_V$, and $D_I$ in all experiments. In Eqn. 2, we set $\lambda = 0.5$ for conventional video generation tasks and use a smaller $\lambda = 0.2$ for cross-domain video generation, as it improves the content consistency. In Eqn. 7, we set $\lambda_m = \lambda_{contr} = \lambda_f = 1$. Grid searching on these hyper-parameters could potentially lead to a performance boost. For TGANv2, we use the released code[4] to train the models on UCF-101 and FaceForensics using 8 Tesla V100 with 16GB of GPU memory.

**Video Prediction.** For video prediction, we predict consecutive frames, given the first frame $\mathbf{x}$ from a test video clip as the input. We find the inverse latent code $\hat{\mathbf{z}}_1$ for $\mathbf{x}_1$ by minimizing the following objective:

$$\hat{\mathbf{z}}_1 = \arg\min_{\hat{\mathbf{z}}_1} \|\mathbf{x}_1 - G_I(\hat{\mathbf{z}}_1)\|_2 + \lambda_{vgg} \|F_{vgg}(\mathbf{x}_1) - F_{vgg}(G_I(\hat{\mathbf{z}}_1))\|_2 , \tag{8}$$

where $\lambda_{vgg}$ is the weight for perceptual loss (Johnson et al., 2016), $F_{vgg}$ is the VGG feature extraction model (Simonyan & Zisserman, 2014). We set $\lambda_{vgg} = 1$ and optimize Eqn. 8 for $20,000$ iterations. We take $\hat{\mathbf{z}}_1$ as the input to our model for video prediction.

---

[2]https://github.com/NVlabs/stylegan2
[3]https://github.com/ajbrock/BigGAN-PyTorch
[4]https://github.com/pfnet-research/tgan2

**AMT Experiments.** We present more details on the AMT experiments for different experimental settings and datasets. For each experiment, we run 5 iterations to get the averaged score.

- *FaceForensics, Ours vs TGANv2.* We randomly select 300 videos from each method and ask users to select the better one from a pair of videos.

- *Sky Time-lapse, Ours vs DTVNet.* We compare our method with DTVNet on the video prediction task. The testing set of Sky Time-lapse dataset includes $2,815$ short video clips. Considering that many of these video clips share similar content and are sampled from 148 long videos, we select 148 short videos with different content for testing. For these videos, we perform inversion (Eqn. 8) on the first frame to get the latent code and generate videos. For DTVNet, we use the first frame directly as input to produce their results. We ask users to chose the one with better video quality from a pair of videos generated by our method and DTVNet. The results shown in Tab. 8 demonstrate the clear advantage of our approach.

Table 8: Human evaluation experiments on Sky Time-lapse dataset.

| Method | Human Preference (%) |
|---|---|
| Ours / DTVNet | **77.3** / 22.7 |

- *FFHQ, Full vs w/o $\mathcal{L}_{\mathrm{contr}}$.* We randomly sample 200 videos generated by each method and ask users to select the more realistic one from a pair of videos.

- *FFHQ, Full vs w/o $\mathcal{L}_{\mathrm{m}}$.* For each method, we use the same content code $\mathbf{z}_1$ to generate 9 videos with different motion trajectories, and organize them into a $3 \times 3$ grid. To conduct AMT experiments, we randomly generate $50\ 3 \times 3$ videos for each method and ask users to choose the one with higher motion diversity from a pair of videos.

**Cross-Domain Video Generation.** We provide more details on the image and video datasets.

- Image Datasets:
  - *FFHQ* (Karras et al., 2019) consists of $70,000$ high-quality face images at $1024 \times 1024$ resolution with considerable variation in terms of age, ethnicity, and background.
  - *AFHQ-Dog* (Choi et al., 2020) contains $5,239$ high-quality dog images at $512 \times 512$ resolution with both training and testing sets.
  - *AnimeFaces* (Branwen, 2019) includes $2,232,462$ anime face images at $512 \times 512$ resolution.
  - *LSUN-Church* (Yu et al., 2015) includes $126,227$ in-the-wild church images at $256 \times 256$ resolution.
- Video Datasets:
  - *VoxCeleb* (Nagrani et al., 2020) consists of $22,496$ short clips of human speech, extracted from interview videos uploaded to YouTube.
  - *TLVDB* (Shih et al., 2013) includes 463 time-lapse videos, covering a wide range of landscapes and cityscapes.

For the video datasets, we randomly select 32 consecutive frames from training videos and select every other frame to form a 16-frame sequence for training.

## C  MORE VIDEO RESULTS

In this section, we provide more qualitative video results generated by our approach. We show the thumbnail from each video in the figures. Full resolution videos are in the supplementary material. We also provide an HTML page to visualize these videos.

**UCF-101.** In Fig. 9, we show videos generated by our approach on the UCF-101 dataset.

**FaceForensics.** In Fig. 10, we show the generated videos for FaceForensics. In Fig. 11 and Fig. 12, we show that our approach can generate long consecutive results, 32 and 64 frames respectively,

even when trained with 16-frame clips. In Fig. 13, we demonstrate that our approach can generate diverse motion patterns using the same content code. In Fig. 14, we apply the same motion codes with different content to get the synthesized videos.

**Sky Time-lapse.** Fig. 15 shows the generated videos for the Sky Time-lapse dataset.

**(FFHQ, VoxCeleb).** Fig. 16, Fig. 17, and Fig. 18 present the generated videos that have motion patterns from VoxCeleb and content from FFHQ, with resolutions of $128 \times 128$, $256 \times 256$, and $1024 \times 1024$, respectively. We use BigGAN as the generator for Fig. 16 and StyleGAN2 for Fig. 17 and Fig. 18.

**(AFHQ-Dog, VoxCeleb).** Fig. 19 presents the generated videos that have motion patterns from VoxCeleb and content from AFHQ-Dog. The videos have a resolution of $512 \times 512$. In Fig. 20, we show the interpolation between every two frames to get longer sequences.

**(AnimeFaces, VoxCeleb).** Fig. 21 shows the generated videos that have motion patterns from Vox-Celeb and content from AmimeFaces. The videos have a resolution of $512 \times 512$.

**(LSUN-Church, TLVDB).** Fig. 22 presents the generated videos that have time-lapse changing style from TLVDB and content from LSUN-Church.

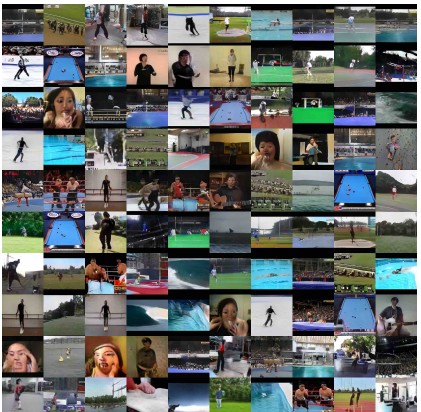

Figure 9: Example videos generated by our approach on the UCF-101 dataset.

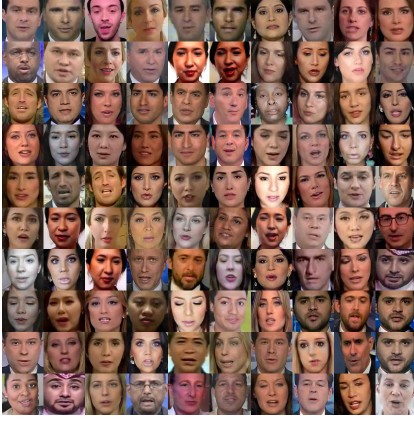

Figure 10: Example videos generated by our approach on the FaceForensics dataset.

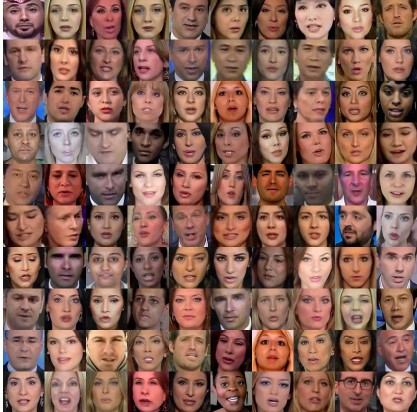

Figure 11: The generated videos on the Face-Forensics dataset consisting of 32 frames.

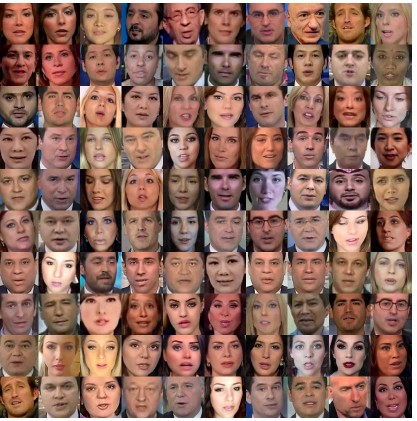

Figure 12: The generated videos on the Face-Forensics dataset consisting of 64 frames.

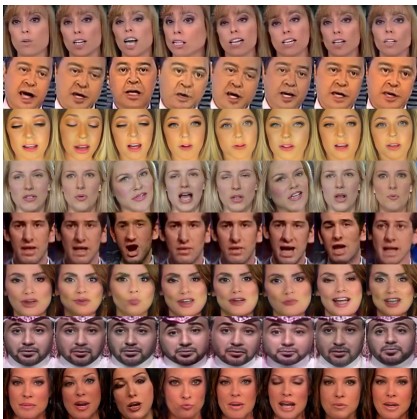

Figure 13: Each row is synthesized using the same content code to generate diverse motion patterns. Please see the corresponding supplementary video for a better illustration.

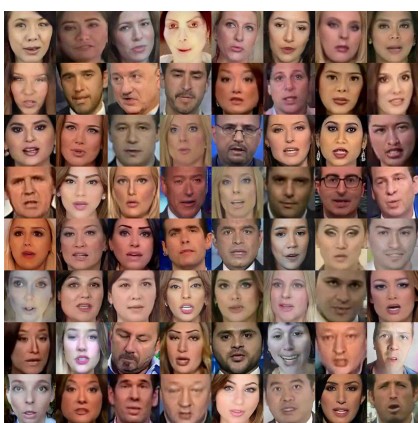

Figure 14: Each row is synthesized with the same motion trajectory but different content codes. Please see the corresponding supplementary video for a better illustration.

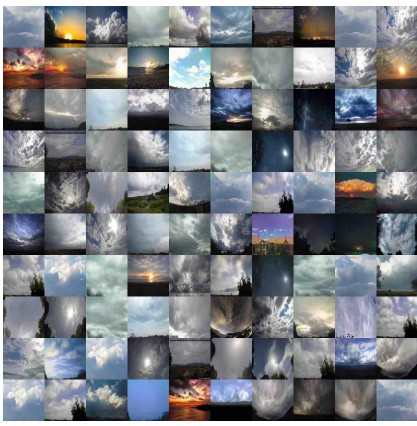

Figure 15: Example videos generated by our approach on the Sky Time-lapse dataset. The videos have a resolution of $128 \times 128$.

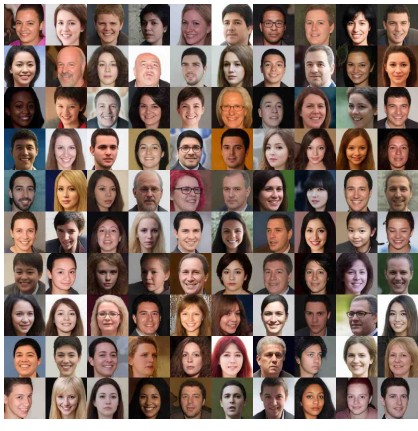

Figure 16: Cross-domain video generation for (FFHQ, Vox). The videos have a resolution of $128 \times 128$.

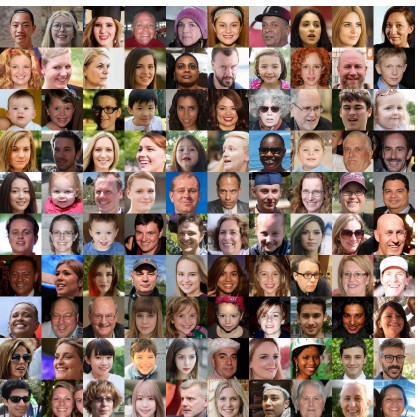

Figure 17: Cross-domain video generation for (FFHQ, Vox). The videos have a resolution of $256 \times 256$.

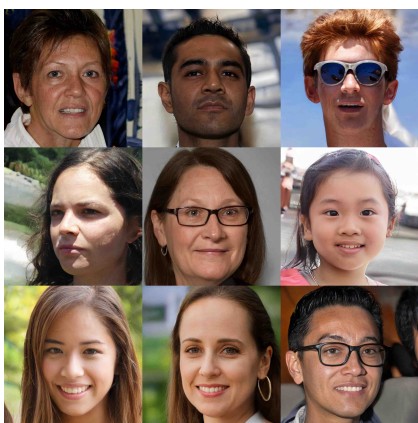

Figure 18: Cross-domain video generation for (FFHQ, Vox). The videos have a resolution of $1024 \times 1024$.

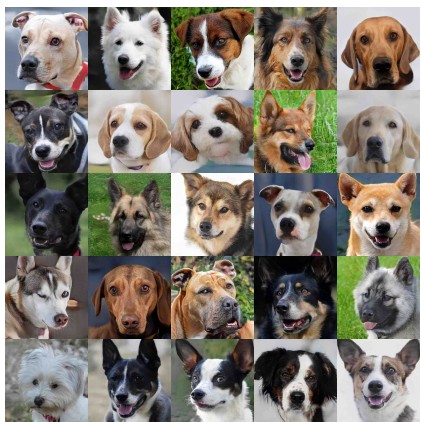

Figure 19: Cross-domain video generation for (AFHQ-Dog, Vox). The videos have a resolution of $512 \times 512$.

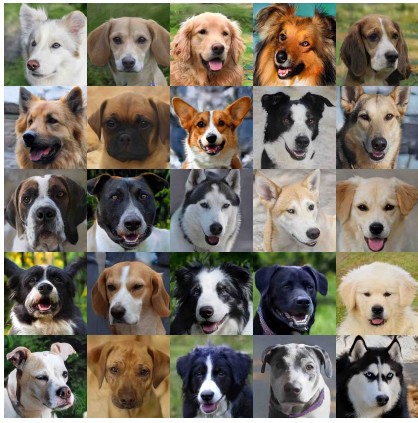

Figure 20: Cross-domain video generation for (AFHQ-Dog, Vox). We interpolate every two frames to get 32 sequential frames. The videos have a resolution of $512 \times 512$.

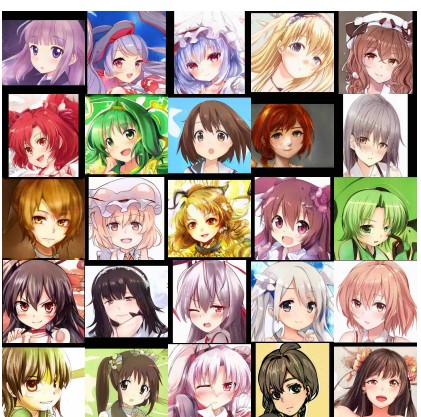

Figure 21: Cross-domain video generation for (AnimeFaces, Vox). The videos have a resolution of $512 \times 512$.

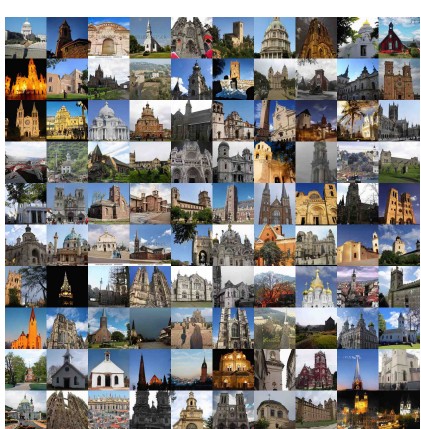

Figure 22: Cross-domain video generation for (LSUN-Church, TLVDB). The videos have a resolution of $256 \times 256$.

## D    MORE ABLATION ANALYSIS FOR MUTUAL INFORMATION LOSS $\mathcal{L}_\mathrm{m}$

In addition to Tab. 5, we perform another ablation experiment to show how mutual information loss $\mathcal{L}_\mathrm{m}$ improves motion diversity by considering the following setting. We random sample a content code $z_1 \in \mathcal{Z}$ and use it as an input to synthesize 100 videos, where each video contains 16 frames. We average the generated 100 videos (they share the same first frame) to get one *mean-video*, which contains 16 frames. For example, for the last frame in the mean-video, it is obtained by averaging all the last frames from the 100 generated videos. We also calculate the per-pixel standard deviation (std) for each averaged frame in the mean-video. More blurry frames and higher per-pixel std indicate the 100 synthetic videos contain more diverse motion.

We evaluate the settings of *Full* and *w/o* $\mathcal{L}_\mathrm{m}$ (without using the mutual information loss) by running the above experiments for 50 times, *e.g.*, sampling $z_1$ for 50 times. Across the 50 trials, for *Full* model, the mean and std of the per-pixel std for the $16^{th}$ frame (the *last* frame in a generated video) is $0.233 \pm 0.036$, which is significantly higher than that of the *w/o* $\mathcal{L}_\mathrm{m}$ model ($0.126 \pm 0.025$). In Fig. 23, we show 8 examples of the last frame from the mean-video and the images with per-pixel std (See supplementary material for the whole videos). Our *Full* model has more diverse motion as the averaged frame is more blurry and the per-pixel std is higher. Note that StyleGAN2 enables noise inputs for extra randomness, we disable it in this ablation study.

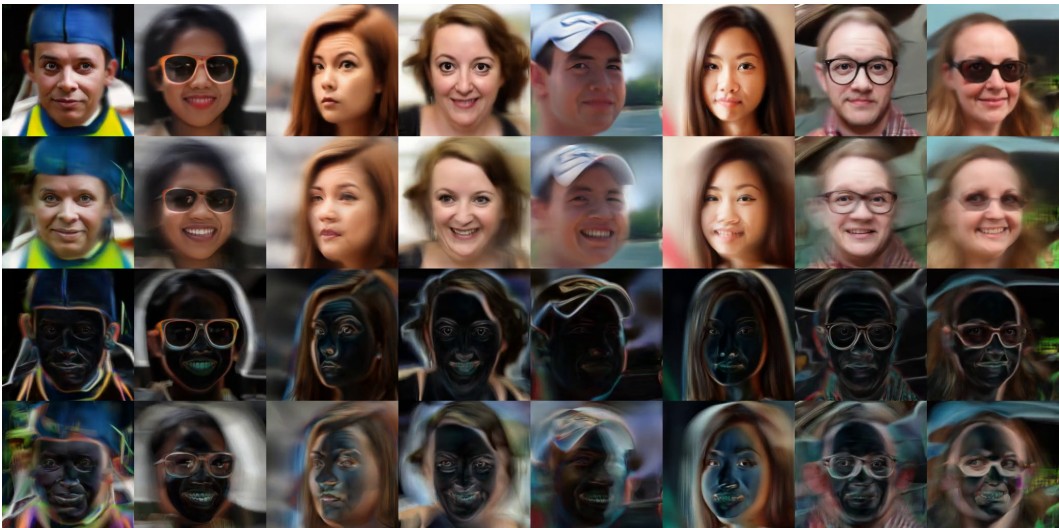

Figure 23: **Row 1 and 3**: The last frame of the mean-video and per-pixel std of *w/o* $\mathcal{L}_\mathrm{m}$ model. **Row 2 and 4**: The last frame of the mean-video and per-pixel std of the *Full* model. The *Full* model has a more blurry mean-video and higher per-pixel std, which indicates more diverse motion.

## E    LIMITATIONS

Our framework requires a well-trained image generator for frame synthesis. In order to synthesize high-quality and temporally coherent videos, an *ideal* image generator should satisfy two requirements: **R1.** The image generator should synthesize high-quality images, otherwise the video discriminator can easily tell the generated videos as the image quality is different from the real videos. **R2.** The image generator should be able to generate diverse image contents to include enough motion modes for sequence modeling.

**Example of R1.** UCF-101 is a challenging dataset even for the training of an *image* generator. In Tab. 7, the StyleGAN2 model trained on UCF-101 has FID $45.63$, which is much worse than the others. We hypothesis the reason is that UCF-101 dataset has many categories, but within each category, it includes relatively a small amount of videos and these videos share very similar content. Such observation is also discussed in DVDGAN (Clark et al., 2019). Although we can achieve

state-of-the-art performance on UCF-101 dataset, the quality of the generated videos is not as good as other datasets (Fig. 9), and the quality of synthesized videos is still not close to real videos.

**Example of R2.** We test our method on BAIR Robot Pushing Dataset (Ebert et al., 2017). We train a $64 \times 64$ StyleGAN2 image generator with using the frames from BAIR videos. The image generator has FID as 6.12. Based on the image generator, we train a video generation model that can synthesize 16 frames. An example of synthesized video is shown in Fig. 24 (more videos are in the supplementary materials). We can see our method can successfully model shadow changing, the robot arm moving, but it struggles to decouple the robot arm from some small objects in the background, which we show analysis follows.

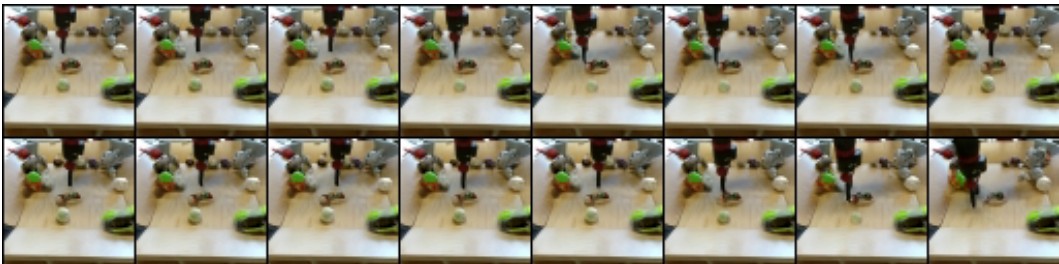

Figure 24: A synthesized video using BAIR dataset. Note the background changing of the first frame (upper-left) and the last frame (bottom-right).

### E.1 ANALYSIS OF THE INFORMATION CONTAINED IN PCA COMPONENTS.

Inspired by previous work (Härkönen et al., 2020), we further investigate the latent space of the image generator by considering the information contained in each PCA component. Fig. 25 shows the percentage of total variance captured by top PCA components. The image generator on BAIR compresses most of the information on a *few* components. Specially, the top 20 PCA components captures 85% of the variance. In contrast, the latent space of the image generator trained on FFHQ (and FFHQ 1024 for high-resolution image synthesis) uses 100 PCA components to capture 85% information. This implies the BAIR generator models the dataset in a low-dimension space, and such generator increases the difficulty for fully disentangling all the objects in images for manipulation.

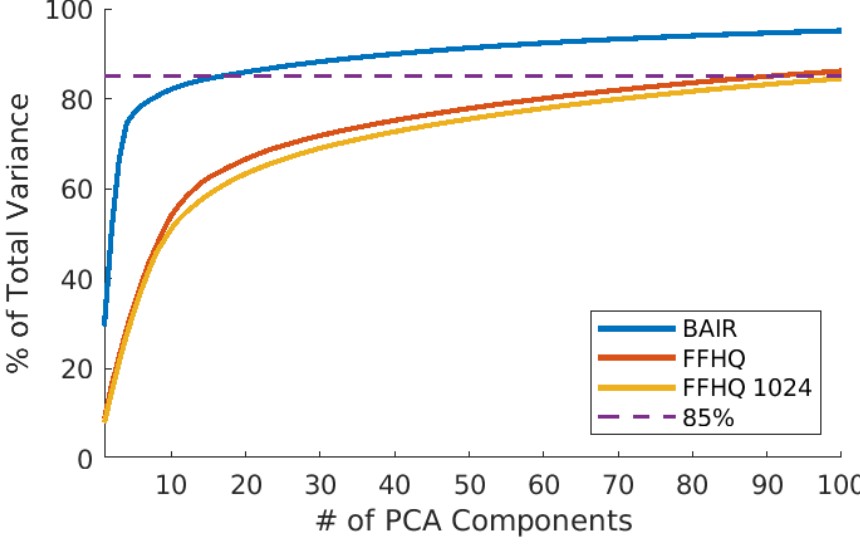

Figure 25: Percentage of variations captured by top PCA components on different models.

Moreover, we visualize the video synthesis results by moving along the top 20 PCA components. Let $V_i$ denote the $i^{th}$ PCA component. Given content code $z_1$, we synthesize a 5-frame video clip by using the following sequence as input: $\{z_1 - 2V_i, z_1 - V_i, z_1, z_1 + V_i, z_1 + 2V_i\}$. In Fig. 26, we show the video synthesis results by moving along the top 20 PCA directions. It can be seen that: 1) changing the later components (the $8^{th}$ and later rows) of BAIR only make small changes; 2) the first 7 components of BAIR have entangled semantic meaning, while the components in FFHQ have more disentangled meaning ($2^{nd}$ row, rotation; $20^{th}$ row, smile). This indicates the image generator of BAIR may not cover enough (disentangled) motion modes, and it might be hard for the motion generator to fully disentangle all the contents and motion with only a few dominating PCA components, while for the image generator trained on FFHQ, it is much easier for disentangling foreground and background.

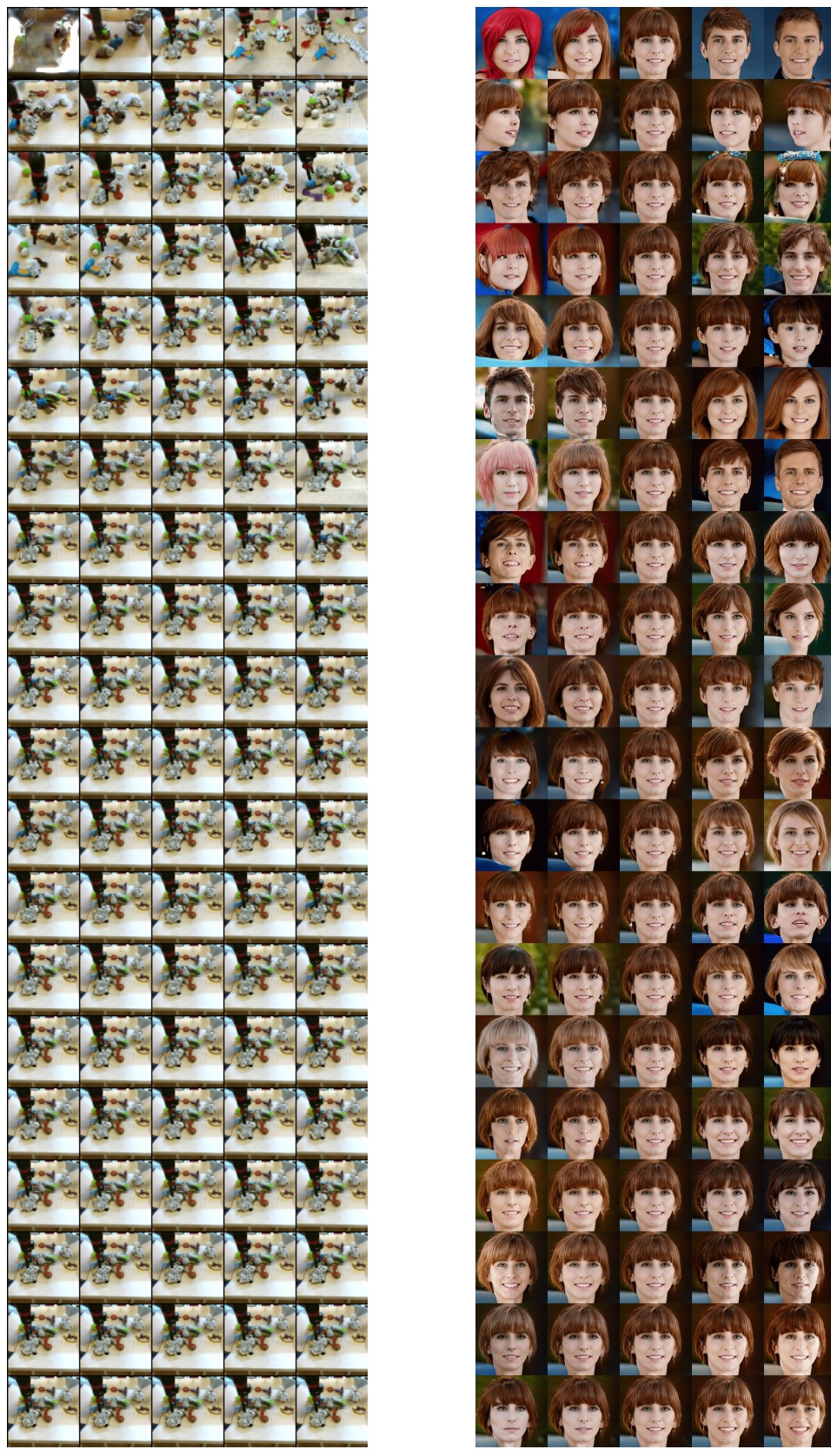

Figure 26: Visualization of top 20 principle components of BAIR (left) and FFHQ (right).

