# OpenReview forum: "A Good Image Generator Is What You Need for High-Resolution Video Synthesis"
_ICLR.cc/2021/Conference — ICLR 2021 Spotlight_

### Official Review · AnonReviewer1 · 2020-10-25
**Review on Paper751**

**Rating:** 6
**Confidence:** 3

**Review:**

Summary

This paper proposes a method to disentangle content and motion from videos for high-resolution video synthesis. The proposed method consists of a motion generator, pre-trained generator, image discriminator, and video discriminator. The motion generator predicts the latent motion trajectory z, which is residually updated over time. Then the image generator produces each individual frame from the motion trajectory. For training, five types of loss functions are combined. In experiments, video generation by the proposed method is performed on UCF-101, FaceForensics, and Sky time-Lapse datasets. Also, cross-domain video generation and more ablation studies were conducted to show the effectiveness of the proposed method.

Overall,  about this paper, I am leaning on the positive side. I summarize both the strength and weakness of this paper that I felt.

Strength

The main benefits of this paper are covering high-resolution video synthesis and disentangling motion and contents. The proposed method was tested on the various datasets and showed experiments about cross-domain video synthesis. Also, ablation studies were performed to check the effects of each loss function.


Weakness


As mentioned in the Introduction, the second desired property for generated video is temporal coherency.  The motion generator may be helpful to find a motion trajectory that makes a temporally consistent video. However, it is unclear whether the meaning of "temporarily constant video" means that the content of the video is consistent over frames in video or there is no flickering effect in the video. Also, there seems to be a lack of experiments to directly verify the temporal coherency.

About the paragraph “Motion Disentanglement” in section 3.1, how to decide variable “m” for PCA? Also, what is the reason why using motion residual is helpful to motion disentanglement?



For evaluation, is there a reason for using different evaluation metrics for each dataset? Also, for each dataset, methods for comparison are different from each other. Is there a special reason to use different methods for comparisons? For, UCF101, IS and FVD were used while FVD and ACD were used for the FaceForensis dataset. Also, FVD, PSNR, and SSIM were used for Sky Time-lapse dataset. It is better to conduct a comparative experiment with the same metric and the same compared methods for all datasets.


There are qualitative results in section4.2 about cross-domain video generation. Are there results that have been verified quantitatively?

---

> ### Author Response · Authors · 2020-11-20
> **Response from the Authors**
>
> We thank the reviewer for the positive feedback and agree that our approach can disentangle motion and content for high-resolution video synthesis and cross-domain video generation, and allows for various experiments on diverse datasets.
>
> **Q1. About temporal consistency.**
>
> **A1.** Here, temporal consistency describes both the consistency of content and motion (*e.g.*, no flickering). The model learns temporal consistency using the video discriminator (using Eqn. 4) and contrastive loss (using Eqn. 6).
>
> As for the evaluation of temporal coherency, we follow the established protocols from previous studies by applying FVD and ACD.  FVD calculates the distance between real and fake video distributions, and thus, it considers whether the synthesized videos have similar temporal consistency as that observed in the real video dataset. We provide the FVD for UCF-101, FaceForesnsics and Sky Time-lapse in Tab. 1, 2, and 3. ACD evaluates facial identity consistency, and we provide the ACD scores in Tab. 2 and 5 for evaluating the facial datasets, *e.g.*, FaceForensics and (FFHQ, VoxCeleb).
>
>
> **Q2. In Section 3.1, how to decide “m” for PCA? Why a motion residual is helpful for the disenglement.**
>
> **A2.** “m” is empirically decided as 384, please refer to Appendix A.1. We do not apply a small value for “m” as we want to retain more informative directions for video synthesis.
>
> The use of a motion residual is inspired by prior efforts in finding interpretable latent directions from generative models [Shen & Zhou, 2020; Harkonen et al., 2020]. These works demonstrate that it is possible to find some interpretable linear directions in the latent space of pre-trained image generators, and these directions contain semantically disentangled properties, such as viewpoint changes and background removal. These works inspire us to disentangle motion with the help of a motion generator. In our ablation study, we have shown that using a motion residual aids in our task (Tab.4, w/o Eqn. 2).
>
> **Q3. Reasons for using different evaluation metrics for different datasets.**
>
> **A3.** The reason is that not all the evaluation metrics are suitable for all the datasets we used in our experiments. Furthermore, we strictly follow evaluation protocols of previous works [Tulyakov et al. 2018, Clark et al., 2019, Zhang et al., 2020] in our evaluations. We explain some details of each metric as follows. Please consult Section 4.1 for further explanation:
> - Inception score (IS) uses a pre-trained C3D model to predict the class probabilities for each generated video. As the C3D model is pre-trained on UCF-101 data, IS is available for UCF-101 evaluation only.
> - ACD evaluates the facial similarity between every two frames within a video, and it uses the pre-trained OpenFace model, thus only works on facial datasets.
> - PSNR/SSIM compares the synthesized video to the ground truth video in pixel/structural level. They are common metrics for video prediction tasks, where we use in the Sky Time-lapse dataset.
> - FVD is a metric that calculates the distance between feature vectors calculated for real and generated videos. It is a generic metric for all kinds of videos, therefore we use it on UCF-101, FaceForensics, and Sky Time-lapse.
>
>
> **Q4. Quantitative results for cross-domain generation.**
>
> **A4.** The existing evaluation metrics are unsuitable for evaluating cross-domain video synthesis results. For example, FVD assumes the generated videos are in the same domain as the real videos, which is not the case for cross-domain video generation, as the two domains have different content. We try our best to evaluate content consistency on (FFHQ, VoxCeleb) with the ACD scores and Human preference experiments via AMT, as shown in Tab. 5. We believe that a new metric is needed for this new research direction. An intuitive solution is to evaluate content and motion separately. For example, the content consistency could be evaluated via similarities between the first and other frames, while the motion consistency could be evaluated by optical flow. However, such a new metric still requires significant experimentation on various datasets to verify its efficacy, which we leave for future work.

---

### Official Review · AnonReviewer3 · 2020-10-27
**You need more than a good image generator**

**Rating:** 8
**Confidence:** 5

**Review:**

Summary: This article proposes to retrofit a pretrained image generator for video generation.

______
Pros:
- Novel objective formulation that controls for motion diversity, disentanglement and content matching.
- High resolution video generation at 1024x1024.
- Cross domain video generation (i.e. use pretrained image generator from one domain on the motion from another).
- Using a pretrained image encoder lowers the training cost and improves efficiency.
- Cost estimates for training the proposed models on commodity hardware are provided.




______
Cons:
- Prior art missing:
  - Proposed approach seems very related to DrNet and a discussion on this is not included (https://papers.nips.cc/paper/7028-unsupervised-learning-of-disentangled-representations-from-video)
  - Proposed approach is missing prior art on efficient video GAN training - (https://www.sciencedirect.com/science/article/abs/pii/S0893608020303397)


_________
Questions:
- Are the LSTM encoder and decoder implemented in practice by the same neural network component?
  - If they are separate components, why not use just one LSTM for both encoding and decoding (the noise vector could be 0 for the first timestep)?
- How will such an approach handle occlusions, during video. A benchmark on the BAIR pushing dataset would be useful to help clarify this.
- "Additionally, each frame in the input video sequence is conditioned on the first frame, as it falls into the distribution of the pretrained image generator, for more stable training."
  - please clarify this statement
- How does this work relate to DrNET?
 - Unsupervised Learning of Disentangled Representations from Video - https://papers.nips.cc/paper/7028-unsupervised-learning-of-disentangled-representations-from-video
 - This is a very related piece of work and it should be included and discussed in the related works section.
 - DrNET seems capable of challenging most the main claims made in this paper.
- How does the proposed approach perform on more complex datasets such as Kinetics-600?
  - It doesn't appear that there is much change in the morphology of video objects temporally, this can be seen on samples from more complex datasets such as UCF-101.
- What is the baseline performance of the model trained on the UCF dataset of videos, if you do not do any temporal modelling but instead tile the first image temporally to make a video ?
  - I suspect that for more complex video with complex motion, there is little in the way of motion modelling happening and the proposed model is simply copying and jittering the initial image.
  - This should be investigated in the paper so as to ascertain the limits of this approach.

_______
Missing References:
-  MDP-GAN: Markov Decision Process for Video Generation - https://openaccess.thecvf.com/content_ICCVW_2019/html/HVU/Yushchenko_Markov_Decision_Process_for_Video_Generation_ICCVW_2019_paper.html
  - Related prior art for video generation
- LDVD-GAN: Lower Dimensional Kernels for Video Discriminators - https://www.sciencedirect.com/science/article/abs/pii/S0893608020303397
  - Related prior art for video generation
  - Relevant to discussion on high resolution video generation as it held the previous state-of-the-art at 512x512 video generation.
  - Table 1 is missing the LDVD-GAN result of 22.91 +/- .19
  - Relevant to the discussion on video GAN efficiency since LDVD-GAN is a single-GPU model.
- DrNET: Unsupervised Learning of Disentangled Representations from Video - https://papers.nips.cc/paper/7028-unsupervised-learning-of-disentangled-representations-from-video
  - Related prior art for video generation
  - Related prior art for proposed disentanglement and training approach.
_______
Overall this work tries to take on video modelling from a different angle with some success but also leaves many questions unanswered due to gaps in the experimental design and related work (discussed above). I will increase my score should these questions and the other concerns be addressed.


__________________________________________________________________________________________________________________________________________
**Updated review** following revisions:
- A5: For future reference, I believe the kinetics dataset is now downloadable from a google drive folder: https://github.com/activitynet/ActivityNet/issues/28#issuecomment-602838701

- A6: I am not convinced by your hypothesis in Appendix (Section E) as to why UCF-101 is so challenging. Surely if UCF-101 is not diverse enough, then a model based on stylegan should perform well on it. In-fact I would argue that it is the opposite, that the UCF-101 is very diverse for such a small dataset.
  - Observe that the datasets compared against UCF-101 in Table 7 are not that diverse, there is a dataset of just faces (FaceForensics), of just sky time-lapses and of just dogs. These are all uni-modal datasets. On the other hand, compare just a subset of the 101 classes in UCF-101; Horse Riding, Military Parade, Baseball Pitch, Billiards Shot, Brushing Teeth,...

  - I would argue that the limitation of your approach (and that of DVDGAN [Clark et al., 2019]) on this dataset stems from the fact that UCF-101 is a small but very diverse dataset. On average, just over 100 samples per class.


Overall, the authors have adequately addressed my questions and concerns. They also appear to have done so for all the other reviewers too.

My recommendation is to **accept** this work for publication to ICLR 2021.

I recommend that the authors open-source their code and pretrained models.

---

> ### Author Response · Authors · 2020-11-20
> **Response from the Authors (part 1/2) - Paper Updated with Suggested References, the Comparison, and More Analysis**
>
> We thank the reviewer for the positive feedback, and agree that we propose a novel formulation to control motion diversity, disentanglement, and content matching, can generate 1024x1024 high-resolution videos, introduce a new direction in cross-domain video generation, and improve the training efficiency with detailed cost estimates provided.
>
> We have updated the manuscript to include the reference papers and comparison with previous work (LDVD-GAN in Tab. 1) as suggested by the reviewer.
>
> **Q1. Are the LSTM encoder and decoder separate components? Why?**
>
> **A1.** We use separate components for the LSTM encoder and decoder, as the inputs for them have different semantic meanings. In our framework, the input for the LSTM encoder is a random code $\mathbf{z}_1$ and ($h=0$, $c=0$). $\mathbf{z}_1$ is sampled from a latent space of pre-trained image generators. The input for the LSTM decoder is a noise vector $\epsilon$ and ($h$, $c$). $\epsilon$ is sampled from a Gaussian distribution and it is used to model motion diversity for the frame, thus it has a different semantic meaning compared to the $\mathbf{z}_1$ used in the encoder.
>
> **Q2. How does the approach handle occlusions? Experiments on the BAIR pushing dataset.**
>
> **A2.** In the paper, we show examples of occlusions. In Fig. 3, we provide examples where clouds moving can reveal or occlude sky. We also portray more examples in Appendix, Fig. 13 (the video with the name Sky-Time-lapse.mp4 in the supplementary material).
>
> We perform additional experiments with the BAIR dataset, for which we first train an image generator, then train a motion generator employing it. We observe that the robot arm motion is somehow entangled with changes to the background in this dataset. Based on our experiments during the limited rebuttal period, we believe that this is because the input latent space of the BAIR image generator is not semantically well-decomposed. However, we believe further investigation is necessary. Please refer to Appendix, Section E of the updated paper for more details. We also show more generated videos for BAIR in the updated supplementary materials (the video named bair.mp4).
>
> **Q3. Clarify this statement: "Additionally, each frame in the input video sequence is conditioned on the first frame, as it falls into the distribution of the pretrained image generator, for more stable training."**
>
> **A3.** Implementation details of the video discriminator are provided in Appendix A.2.1. Here we provide more explanation of the design. An observation is that, as our image generator is first pre-trained and fixed during the training of video generation, the first frames of the synthesized videos are always high-quality (they fall into the distribution of the pre-trained image generator). Based on this observation, conditioning videos on the first frame provides more information to the discriminator, such as the frame quality consistency and the motion changing. A model without such a condition has IS 31.98 on UCF-101, which is lower than the model with this condition (IS: 33.95).

---

> > ### Author Response · Authors · 2020-11-20
> > **Response from the Authors (part 2/2) - Paper Updated with Suggested References, the Comparison, and More Analysis**
> >
> > **Q4. Comparison with DrNET.**
> >
> > **A4.** We thank the reviewer for pointing out this paper. After careful inspection, we find that it focuses on learning disentangled image representations from video, and they show its effectiveness in video prediction/classification tasks. Our work mainly focuses on video generation, a different task. Overall, we believe that our work is not in conflict with DrNET. In particular, we note several distinct differences:
> >
> > First, DrNET is not a generative model. Their adversarial loss is not applied to the decoder. As a result, to perform video prediction, their LSTM requires initial *input coming from the encoders*. On the other side, our generative model is trained with adversarial loss, and the LSTM thus takes a *random noise vector* $\mathbf{z}_1$ as input, which is sampled from a Gaussian distribution.
> >
> > Second, We note that the general concept of learning to disentangle motion from content has been explored in other works, *e.g.*, those cited in the paper, including some roughly contemporary to DrNET such as [Tulyakov et al. 2018].  We believe that our general approach differs drastically from these works in ways that present several other key advantages. For example, as DrNET *concatenates* the content and pose when synthesizing images, which is an approach not employed for state-of-the-art image generator training, such generators would have to be adapted and then re-trained to be used with this technique. Instead, we design our content/motion disentanglement in an *additive* manner, learning the appropriate sequence of images to sample from an off-the-shelf image generator to synthesize coherent videos. This allows us to exploit the benefits of state-of-the-art generators such as StyleGAN, *e.g.*, high image resolutions and quality that are not presently possible when learning to directly synthesize videos. It also allows for other applications, such as the cross-domain video synthesis provided in our evaluations.
> >
> > Finally, other major differences include that 1) we synthesize high-resolution videos, while DrNET is not, and 2) we use pre-trained image generators, while DrNET is not.
> >
> > If the reviewer has other questions or comments regarding the differences between DrNET and our work, we are happy to answer or address them.
> >
> > **Q5. About Kinetics-600.**
> >
> > **A5.** While working on the paper, we made multiple attempts to evaluate our work on Kinetics-600. Unfortunately, we realize that YouTube does not allow an IP address to download videos in parallel, and it also has other limits preventing obtaining such data. Given a single external IP address the estimated download time is more than 5 months. We are still downloading the dataset. Furthermore, we find some video links provided in Kinetics-600 are not available any more on YouTube, which means we can not perform a fair comparison even if we download all the available videos. We also find multiple discussions on the web in which other researchers share similar concerns. However, we would eagerly run the necessary experiments should there exist a method to get such data of which we are not aware of.
> >
> > **Q6. Performance on UCF-101 by tiling the first image temporally to make a video.**
> >
> > **A6.** The IS and FVD for the first-frame-tiled video on UCF-101 is 27 and 1400, respectively, which are worse than existing state-of-the-art works.  With our method, we can improve the performance and achieve much better results (IS: 33.95 and FID: 700). For UCF-101, we do observe limitations of our approach. As our method is based on the pre-trained image generator, the quality of the generator can affect the performance of video generation. We find UCF-101 is a challenging dataset even for image generation, which is also observed by DVDGAN [Clark et al., 2019]. Therefore, modeling motion from a not-well trained image generator becomes hard. We also provide a new section in Appendix (Section E) to analyze this. Please refer to that section for more details.

---

### Official Review · AnonReviewer2 · 2020-10-28
**Interesting approach and extensive experiments**

**Rating:** 8
**Confidence:** 5

**Review:**

**Summary**
This paper addresses the problem of video synthesis --- generating diverse, realistic videos. This paper's core idea is to leverage a fixed, pre-trained GAN model for image synthesis and train a motion generator to produce a sequence of latent vectors to generate image sequences (using the pretrained GAN and the generated latent vectors) are temporally coherent. The specific technical novelties lie in (1) predicting the motion residual and (2) adding contrastive image discriminator to ensure that generated contents in a video are similar. The paper provides an extensive set of experiments demonstrating the proposed method's effectiveness over the state-of-the-art video synthesis models.

**Strength**
+ The quantitative and visual results are extensive. The performance over existing methods across multiple datasets is significant. The paper also provides an ablation study (Table 4, 5) highlighting the importance of individual components.
+ The capability of cross-domain video synthesis could be beneficial, particularly when high-quality video datasets are difficult to collect.
+ The appendix and the supplementary material provide extensive details about the experimental settings and results that would help reproduce the results.


**Weakness**
- When discussing the difference over [Tulyakov et al. 2018], the paper states “…applies h_t as the motion code for the frame to be generated, while the content code is fixed for all frames. However, such a design requires a recurrent network to estimate the motion while preserving consistent content from the latent vector, … difficult to learn in practice”. I do not fully understand why this is the case. It would be clearer if the paper can explain why such a design causes difficulty in learning and why the proposed design could alleviate such problems.

- For motion diversity, why maximizing the mutual information between the hidden vector and the noise vector can prevent mode collapse?

- It seems to me that the proposed method can only handle 1) “subtle” motion, such as facial expressions and 2) short video sequences (e.g., 16 frames). One can see the problem in the synthesized results for UCF-101: inconsistent motion, changing color, or object disappearing over time. It would be interesting to videos with a longer duration (by running the LSTM over many time steps).

In sum, this is a paper with an interesting idea and extensive experiments. While the results are still not perfect and seem to handle subtle motion, the quantitative and qualitative evaluation show clearly improved results over the previous state-of-the-art.

---

> ### Author Response · Authors · 2020-11-20
> **Response from the Authors**
>
> We thank the reviewer for the positive feedback and agree that 1) our approach is novel for video generation, and has achieved significant improvements over existing studies with extensive quantitative and qualitative experiments, and 2) the cross-domain video generation could be beneficial for high-resolution video synthesis.
>
> **Q1. More details for why MoCoGAN’s [Tulyakov et al. 2018] design is difficult to learn in practice.**
>
> **A1.** The input to MoCoGAN is the concatenation of content codes sampled from a normal distribution and motion codes sampled from a recurrent network. Their framework learns to disentangle content and motion by using video data as a supervision. While in our method, content is learned using image data, and motion is learned from videos. We train the image generator with image data only. Our design can find semantically meaningful directions from the input codes and add motion to the already learned content.
>
> **Q2. For motion diversity, why maximizing the mutual information between the hidden vector and the noise vector can prevent mode collapse?**
>
> **A2.** We generate videos by sampling random noise vectors. These vectors are diverse since they are independently and identically distributed. Achieving similar diversity in the output of the video generator is a challenging task. As we mentioned in Section 3.1 (paragraph Motion Diversity): we observe the LSTM decoder tends to neglect the motion information (random noise vectors $\epsilon$), which leads to near deterministic generation (we also observe a “temporal mode collapse” in prior works, in MoCoGAN [Tulyakov et al. 2018], within the motion category, *e.g.,* for fear, the facial expressions are the same for different people (https://github.com/sergeytulyakov/mocogan)). Therefore, we use the mutual information loss to ensure that the LSTM decoder uses the input motion information $\epsilon$, and attempts to synthesize diverse outputs. Please refer to the video files with the names FFHQ_with_mutual_loss.mp4, which contains the mutual information loss, and FFHQ_without_mutual_loss.mp4, which does not contain the mutual information loss, in the supplementary materials for more visual comparisons and results.
>
> Additionally, in the updated paper, Appendix, Section D, we introduce more ablation experiments for the motion diversity by quantitatively evaluating the pixel-wise difference between generated videos with and without the mutual information loss.
>
> **Q3. About “subtle” motion and long videos.**
>
> **A3.** In addition to human, dog, and anime facial expression, our method can also model timelapse changing and sky time-lapse motions, which involve significant changes in lighting and texture caused by different time of the day. We believe these are not subtle motions. For example, we illustrate that the sky can be visible or occluded when clouds are moving in Fig. 3 and 13 (the video with name sky-time-lapse.mp4 in the supplementary material).
> The motion inconsistency problem for UCF-101 is due to the image generator not being well-trained on this data, as we find that the UCF-101 dataset is very challenging, even for image generation. In Tab. 7 (Appendix B), we show the FID score for the UCF-101 dataset is 45, which is worse than other datasets, *e.g.,* FID as 10.99,10.80, and 7.85 for FaceForensics, Sky Time-lapse, and AFHQ-Dog respectively. When the image generator is not well trained, it can affect the motion generator training, especially for video generation, as the video discriminator can easily tell the differences between synthetic and real videos. We add the analysis of UCF-101 in Appendix, Section E (highlighted in blue).
>
> Our method can generate longer videos than it has seen during training as shown in Appendix Fig. 9, 10, and 18. To do that, our framework offers two possibilities:
> - We can run the LSTM Decoder for more steps (as mentioned by the reviewer). We show examples of 32-frame and 64-frame generation in Fig. 9 and 10 (video files with the names FaceForensics-32.mp4 and FaceForensics-64.mp4 in the supplementary materials).
> - We can do interpolation on the motion trajectory directly to synthesize long videos. We show examples of 32-frame generation in Fig. 18 for cross-domain video generation (the video file with the name AFHQ-DOG-Interpolate_32.mp4 in the supplementary materials).
>
> Generating long videos, such as a couple of minutes long, with rich dynamics remains challenging and is future work. Data, memory, compute time, and cost being the main limiting factors here.

---

### Official Review · AnonReviewer4 · 2020-10-29
**Interesting idea decent results**

**Rating:** 6
**Confidence:** 2

**Review:**

An interested outsider trying to exercise best judgment here.

Certainly interesting idea with great results. The way this paper approaches video synthesis is to decouple that into two steps: 1) learning motion (residuals between the latent codes of consecutive frames) from a sequence of past motions, and 2) synthesizing a new image using the learnt latent code. Would you agree with this layman's assessment?

Reasons to accept the paper:
* The decoupling idea is new and interesting from an interested outsider's perspective.
* Evaluation is pretty comprehensive and the results are decent.

I have a few clarification questions though.
* The related work section claims that "In this paper, we focus on generating realistic videos using manageable computational resources" Do you have results to show that your approach is computationally more efficient?
* Intuitively, why do you think future video frames can be predicted? That is, fundamentally, why do you think motion can be learnt?If I understand you correctly, you basically claim that given a sequence of motion from the training data, we should be able to learn the subsequent motion. Isn't it that there could be many kinds of subsequent motion given a sequence of past motions? I could understand how you are using it in the context of image synthesis, i.e., as long as the predicted motion can be used to generate realistic images you'd be fine with it, but precisely predicting the next frames seems to suggest that there is one fixed motion given the past motion sequence.
* The whole motion diversity thing feels quite hand-wavy and a bit empirical to me. It feels like a hack. Why is it that motion needs to be diverse? In fact, how could you precisely quantify diverse?
* Equation 4 (and the last paragraph of Section 3.1) shows that the motion generation is evaluated with the loss using the image generated from the image generator. Do you have a more direct way of quantifying the quality of the learnt motion? Very minor though.

---

> ### Author Response · Authors · 2020-11-20
> **Response from the Authors**
>
> We thank the reviewer for the positive feedback, and agree that our paper certainly introduces interesting ideas with great results validated by comprehensive experiments.
>
> **Q1. How motion is learned, and why motion can be predicted.**
>
> **A1.** We would like to emphasize that video generation and video prediction are two different problems with different settings. For video prediction, the goal is to predict future frames in a video given the observed frames in the video, while for video generation, no previous frame is given.
>
> In this work, we mainly focus on video generation, and thus our goal is to synthesize videos that are realistic. Therefore, we do not extract or estimate motion from previous frames (past motion). Instead, the motion is learned during the training stage. The motion generator and the video discriminator are the key components that perform this task. Specifically, for training, we pass sampled motion trajectories (random noise) to the motion generator for video synthesis, and the discriminator is trained to distinguish between real and synthetic videos (more details are in Section 3.1).
>
> In video prediction, future video frames are predicted given previous frames. For example, if we see a man walking from left to the center within a few frames, then, in the next frame, it is highly likely this man will keep walking to the right. However, as the future is not deterministic, there is a possibility that the man will turn around suddenly or perform a different action.
>
> **Q2. About the results showing the proposed approach is more computationally efficient.**
>
> **A2.** In the paper, we compare our computational cost with the existing state-of-the-art video generation model, DVDGAN, on Google Cloud. We estimate the training cost for DVDGAN is more than *$30K* (Page 1, footnote). While the training cost of our method for all the datasets that we used ranges from *$0.7K*~*$2.3K* (Appendix Section B, in the paragraph Training Time). Our training cost is thus much lower than that of DVDGAN, and we note that Reviewer 3 suggests that part of our contribution is the reduction of this cost, improving the overall efficiency and providing cost estimation on commodity hardware.
>
> **Q3: Why does motion need to be diverse? How could you precisely quantify diversity?**
>
> **A3.** Motion needs to be diverse as the same action, such as walking to the left, smiling, etc can be performed differently, at different pace, starting from a different initial state and so on. Therefore, to adequately model videos, video generation frameworks should be able to generate diverse motions. We qualitatively show such diversity in Fig. 5, Fig. 11 and in FaceForensics_SameCont_DiffMotion.mp4 in the supplementary material.
>
> In Tab. 5, we provide human evaluation results to show that the mutual information loss (Eqn. 3) improves the motion diversity quantitatively. Additionally, we include a new section in the Appendix (Section D) to quantitatively evaluate the pixel-wise difference between the generated videos with and without the mutual information loss. Please refer to that section for more details.
>
> **Q4. Eqn. 4 shows motion is evaluated with the loss by leveraging an image generator. A more direct way of quantifying the quality of learned motion.**
>
> **A4.** Eqn. 4 is an adversarial loss describing the way we learn motion by using real videos. It is not related to our evaluations. To quantitatively evaluate the quality (content and motion) of the generated videos, we use a widely-adopted metric by the community, which is the FVD. To our knowledge, there is no similarly widely-adopted and agreed-upon metric for motion quality that would apply to the motion of arbitrary objects and scenes, and for which the video content would not have as significant an impact on the results. However, as the FVD is designed to work on videos, the quality of the motion is a significant part of this metric, and thus we decide to employ it for our evaluation of this aspect of our results. In Tab. 1, 2, and 3, we have compared our approach with prior state-of-the-art approaches on three datasets using FVD. Results show that our method achieves the best video generation results.

---

### Decision · Program_Chairs · 2021-01-07
**Final Decision**

**Decision:**

Accept (Spotlight)

**Comment:**

All the reviewers are positive about the paper; R2 and R3 voted for clear accept. Overall, all the reviewers feel that evolution is comprehensive and the results are decent. There is a novel objective formulation that controls for motion diversity, disentanglement and content matching, outperforming existing methods across multiple datasets. High-res videos at 1024x1024 are generated and there is cross-domain video generation. Many good questions were raised by the reviewers, and they were addressed in details in the rebuttal. In particular, the question about subtle motion and short video sequences was raised (which was the concern that the AC had). The AC agrees with the reviewers that the paper warrants a publication. Please address the questions raised by the reviewers in the final version.